# Stat3 regulates centrosome clustering in cancer cells via Stathmin/PLK1

Edward J. Morris[1], Eiko Kawamura[1,†], Jordan A. Gillespie[1], Aruna Balgi[2], Nagarajan Kannan[3], William J. Muller[4], Michel Roberge[2] & Shoukat Dedhar[1,2]

Cancer cells frequently have amplified centrosomes that must be clustered together to form a bipolar mitotic spindle, and targeting centrosome clustering is considered a promising therapeutic strategy. A high-content chemical screen for inhibitors of centrosome clustering identified Stattic, a Stat3 inhibitor. Stat3 depletion and inhibition in cancer cell lines and in tumours *in vivo* caused significant inhibition of centrosome clustering and viability. Here we describe a transcription-independent mechanism for Stat3-mediated centrosome clustering that involves Stathmin, a Stat3 interactor involved in microtubule depolymerization, and the mitotic kinase PLK1. Furthermore, PLK4-driven centrosome amplified breast tumour cells are highly sensitive to Stat3 inhibitors. We have identified an unexpected role of Stat3 in the regulation of centrosome clustering, and this role of Stat3 may be critical in identifying tumours that are sensitive to Stat3 inhibitors.

[1] Department of Integrative Oncology, BC Cancer Research Centre, BC Cancer Agency, Vancouver, British Columbia, Canada V5Z 1L3. [2] Department of Biochemistry and Molecular Biology, Life Sciences Institute, University of British Columbia, Vancouver, British Columbia, Canada V6E 4A2. [3] Terry Fox Laboratory, BC Cancer Agency, Vancouver, British Columbia, Canada V5Z 1L3. [4] Department of Biochemistry, Rosalind and Morris Goodman Cancer Centre, McGill University, Montreal, Quebec, Canada H3A 1A3. † Present address: Western College of Veterinary Medicine, University of Saskatchewan, Saskatoon, Saskatchewan, Canada S7N 5B4. Correspondence and requests for materials should be addressed to E.J.M. (email: emorris@bccrc.ca) or to S.D. (email: sdedhar@bccrc.ca).

In many types of cancers, centrosome amplification is observed at a high frequency and is associated with poor patient outcomes[1–5]. Centrosome amplification is thought to be caused by both faulty, incomplete mitosis and overexpression of genes involved in centrosome duplication[6]. The significance of centrosome amplification in cancer is not fully understood, although cancer cell lines with supernumerary centrosomes are more invasive[7], supporting the hypothesis that centrosome amplification has a role in cancer progression and metastasis. Tumours with supernumerary centrosomes have greater levels of chromosome missegregation and aneuploidy, suggesting that centrosome amplification might have a role in increasing mutation rates and therefore cancer progression[8]. While induction of centrosome amplification leads to tumour formation in Drosophila[9] and transient centrosome amplification promotes formation of skin tumours in mice[10,11], chronic centrosome amplification did not increase the rate of cancer initiation in mice engineered to overexpress a centrosome amplification gene[12,13].

During cell division, the two centrosomes can usually position themselves at the ends of the mitotic spindle, but it appears that when there are supernumerary centrosomes, additional mechanisms are required to cluster the centrosomes together to form a bipolar spindle. For instance, cortical actin and microtubule motors pull on astral microtubules to cluster the excess centrosomes together[14,15].

Inhibiting supernumerary centrosome clustering in mitosis is an attractive strategy to target cancer cells while sparing normal cells, since supernumerary centrosomes are almost exclusively found in cancer cells and inhibition of centrosome clustering induces mitotic defects and cell death[6]. We previously performed a phenotypic screen for inhibitors of centrosome clustering using a library of compounds with drug-like properties[16]. We have greatly expanded this screen with 10,000 new compounds and have identified a compound KM08165 that is effective at reducing the viability of cancer cells with centrosome amplification while being significantly less toxic to normal cells without centrosome amplification. We tested chemical substructures of KM08165 and discovered that the Stat3 inhibitor Stattic, a predicted degradation product of KM08165, is a potent inhibitor of centrosome clustering. Here we elucidate a pathway involving Stat3, Stathmin and PLK1, which regulates γ-tubulin levels at the centrosome to allow supernumerary centrosomes to correctly position themselves and become clustered.

Stat3 is frequently overexpressed in cancer and has been implicated as a critical factor in cancer progression, acting as a transcription factor for growth promoting and anti-apoptotic genes[17]. We demonstrate that Stat3 is also involved in the regulation of supernumerary centrosome clustering, revealing a new function for a critical cancer-related gene.

## Results

**Centrosome clustering inhibitor screen identifies Stattic**. An automated phenotypic screen for the identification of compounds that inhibit cancer cell centrosome clustering was conducted (Supplementary Fig. 1a). Compounds from an extended Maybridge chemical library were scored as hits if the compound-treated cells had an increase in the percentage of mitotic cells with more than two distinct centrosomes, indicating declustering[16]. Out of the 10,000 compounds tested, the best 'hit' was compound KM08165 (Fig. 1a–c; Supplementary Fig. 1b–d). Further characterization of this compound showed that it was a promising anti-cancer candidate because it was more effective in reducing cell viability in several cancer cell lines versus non-tumorigenic cell lines, freshly isolated normal primary human mammary epithelial cells (HMECs) from reduction mammoplasties or normal primary human bone marrow cells. The cell lines (BT-549, RPMI-8226 and OPM-2) were chosen because they have been previously shown to be highly sensitive to centrosome clustering inhibitors[18], and the multiple myeloma cell lines RPMI-8226 and OPM-2 have a high centrosome index[19] that make them good candidates to test centrosome clustering inhibitor treatments.

To identify potential biological targets, we carried out structure–function analysis of substructures[20] of KM08165 (Fig. 1d), measuring effectiveness at inhibiting centrosome clustering and ability to reduce cell viability. Out of the compounds tested, Stattic was more effective than the parental KM08165 at inhibiting centrosome clustering and equally effective at reducing cell viability in cancer cells (Fig. 1e–g; Supplementary Fig. 1e,f). Pilot studies using an established liver-derived microsome assay and ultra-violet ultra-performance liquid chromatography (UPLC-UV) detection suggested KM08165 is metabolically converted into Stattic (Supplementary Fig. 1g), suggesting that Stattic is a metabolic degradation product of KM08165. Since Stattic is a Stat3 inhibitor[21], we hypothesized that Stat3 regulates centrosome clustering in cancer cells. A schematic of the inhibitor treatment timeline used in this paper is shown in Supplementary Fig. 1h.

**Stat3 is required for centrosome clustering**. To test whether Stat3 regulates centrosome clustering, we treated BT-549 and MDA-MB-231 human breast cancer cells with Stat3 short interfering RNAs (siRNAs) and counted the number of mitotic cells with declustered centrosomes. Three out of four Stat3 siRNAs tested inhibited centrosome clustering and the percentage of cells with declustered centrosomes corresponded with the effectiveness of the Stat3 siRNA (Fig. 2a; Supplementary Fig. 2a). In support of this, other Stat3 inhibitors with differing mechanisms of action also induced concentration-dependent increases in centrosome declustering (Fig. 2b; Supplementary Fig. 2b,c).

A change in the number of separated centrosomes per cell could be due to centrosome declustering but could also be due to centrosome amplification or fragmentation. To determine whether the effects of Stat3 inhibition are due to inhibition of centrosome clustering, or related phenomena, we immuno-stained mitotic BT-549 cells for centrin and counted centriole pairs. Stattic (1 μM) induced all the BT-549 cells with supernumerary centrosomes to become declustered but did not change the number of centriole pairs, suggesting that centrosome amplification is not occurring (Fig. 2c). We also quantified the percentage of Stattic-treated cells with centrosome fragmentation, using γ-tubulin and centrin-2 staining (Fig. 2d). The percentage of mitotic BT-549 cells with non-centrin-associated γ-tubulin was the same in untreated and Stattic-treated cells, showing that 1 μM of Stattic does not induce fragmentation, only declustering (Fig. 2d).

While pericentrin is frequently used to visualize centrosomes, it is not a specific marker of centrioles. Therefore, to confirm that Pericentrin staining is a reliable marker for scoring centrosome clustering, we scored centrosome clustering using both Pericentrin and Centrin-2 (Supplementary Fig. 2d,e). The number of cells with declustered centrosomes was identical when the cells were counted using anti-pericentrin or anti-centrin-2 antibody staining, both in untreated and Stattic-treated cells. This observation suggests that scoring centrosome clustering with anti-pericentrin is as reliable as scoring with antibodies to centrin.

For in vivo confirmation, we scored the percentage of mitotic cells with declustered centrosomes in breast-specific wild-type (WT) and Stat3 − / − mouse tumours[22]. While the frequencies of mitotic cells with supernumerary centrosomes were similar between WT and Stat3 − / − tumours, mitotic cells in Stat3 − / − tumours had a significantly higher proportion of declustered

centrosomes compared to the WT tumours, demonstrating that Stat3 regulates centrosome clustering in tumours *in vivo* (Fig. 2e; Supplementary Fig. 2f).

Previous work suggested that Stat3 promotes centrosome amplification[23]. However, we did not observe Stat3-dependent changes in the number of cells with supernumerary centrosomes

*in vitro* (Fig. 2c) or *in vivo* (Fig. 2e). We repeated the published experiments and additionally tested Stat3 inhibitors Stattic and BBI-608 using the same conditions (Supplementary Fig. 2g,h). As reported, the Stat3 inhibitor piceatannol inhibited hydroxyurea-induced centrosome amplification in CHO cells; however, Stattic and BBI-608 had no effect. Piceatannol is now

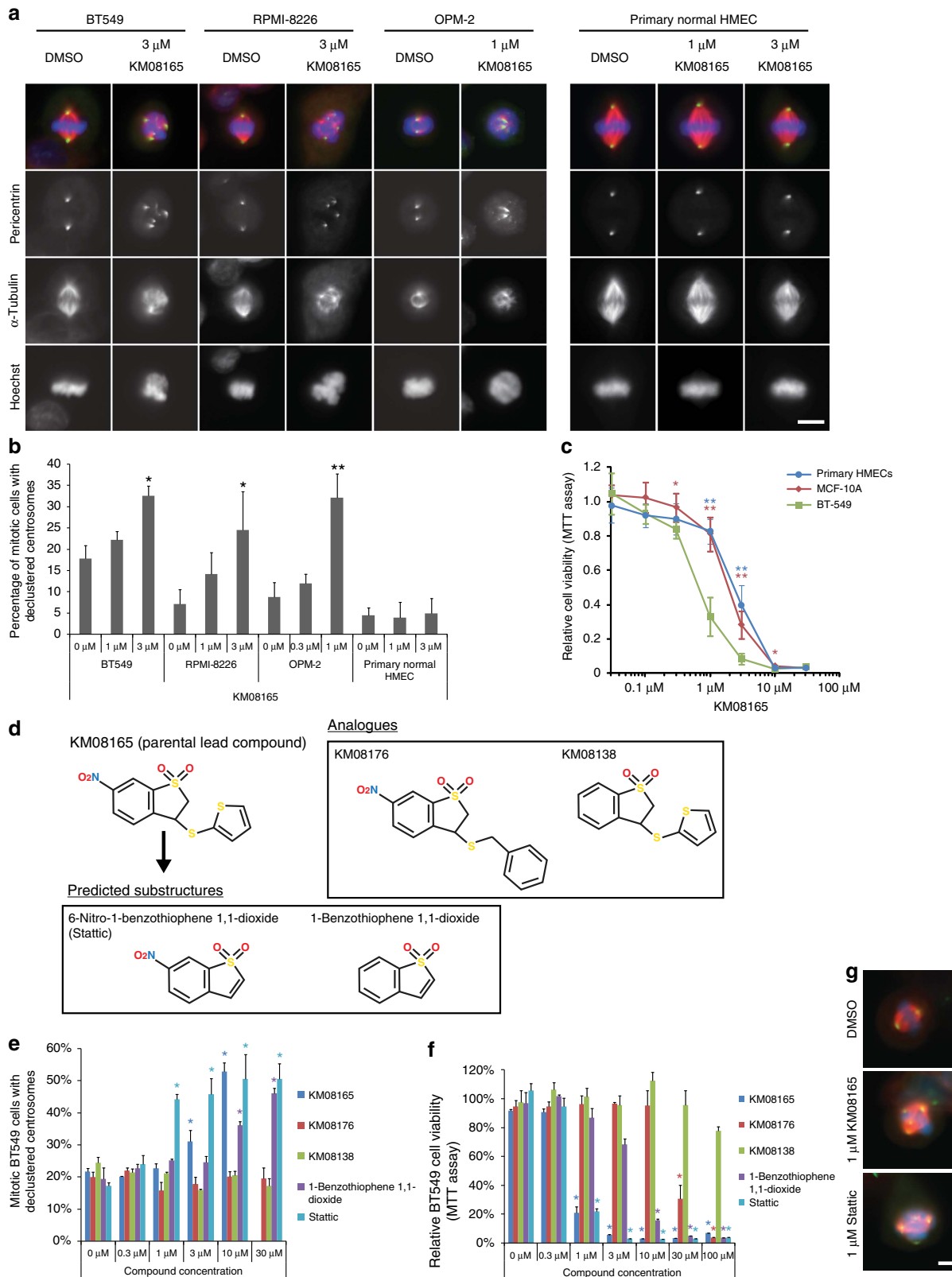

known to be a broad range inhibitor that primarily affects the mitotic kinase Syk[24]. We speculate that inhibition of centrosome amplification by piceatannol is due to Stat3-independent effects, possibly via inhibition of Syk, since Syk was shown to promote centrosome amplification[25].

**Stat3–centrosome clustering is transcription-independent**. Stat3 is widely acknowledged as a regulator of gene transcription. Thus, to determine whether Stat3 regulates centrosome clustering in a transcription-dependent manner, we scored centrosome clustering in the presence of the messenger RNA synthesis inhibitor Actinomycin D and the protein synthesis inhibitor cycloheximide with and without Stattic (Fig. 3a). Neither actinomycin D nor cycloheximide inhibited centrosome clustering, and Stattic was effective at inhibiting centrosome clustering in the presence of actinomycin D and cycloheximide. To confirm that Stattic, cycloheximide and actinomycin D were effective at blocking transcription and/or protein synthesis, we employed a Stat3-specific reporter assay (Fig. 3b) and found that these inhibitors were able to block transcription or translation, in line with previous studies[26]. Cycloheximide and actinomycin D treatments have been previously used in mitotic cell experiments where they blocked entry into S-phase, slowed mitotic progression and blocked completion of cytokinesis; however, besides that mitosis proceeded unimpaired[27–31]. We found a small decrease in the number of cells in mitosis with cycloheximide (from 3.6% for untreated cells to 2.9% for 30 µM cycloheximide) as well as for actinomycin D (from 3.9% for untreated cells to 3.4% for 30 µM actinomycin D). One possible explanation for this relatively small difference is that while cells are being blocked from entering mitosis, they are also being prevented from exiting anaphase and so the two processes roughly equal out.

Next, we explored how Stat3-dependent mitotic centrosome clustering is regulated. In canonical Stat3 pathways, Stat3 is activated by Jak2, which is in turn activated by receptors such as the epidermal growth factor receptor (EGFR)[32,33] (Supplementary Fig. 3). We found that centrosome clustering is not affected by epidermal growth factor activation of EGFR (Fig. 3c, left), EGFR inhibition by erlotinib (Fig. 3c, right) or Jak2 inhibition by ruxolitinib (Fig. 3d), suggesting that Stat3 regulation of centrosome clustering is independent of canonical Stat3 upstream signalling. One of the primary ways Stat3 is activated is by inducing shuttling of Stat3 into the nucleus; however, since we are examining mitotic cells and the nucleus is not present in mitotic cells, it seems likely that the regulation of Stat3 is different in our experiments. Stat3 has not been previously studied in mitosis however in interphase, Stat3 is primarily regulated through phosphorylation[34]. To examine this aspect of Stat3 regulation, we generated stable cell lines that express constitutively active (Stat3C) and kinase-dead (Stat3-Y705F)

mutants then knocked down endogenous Stat3 with siRNA. Stat3C increased the number of cells with clustered centrosomes as well as rescued Stat3 siRNA-treated cells (Fig. 3e,f), which suggests that this active form of Stat3 regulates centrosome clustering. Stat3-Y705F cannot be tyrosine phosphorylated and is primarily monomeric[35]. We found that Stat3-Y705F blocked centrosome clustering and could not rescue Stat3 siRNA knockdown. Overall, these results suggest that the active form of Stat3 is the primary mediator of mitotic centrosome clustering.

**Stathmin is downstream of Stat3 in centrosome clustering**. Since centrosome clustering is independent of Stat3 transcriptional factor function, we examined whether Stat3 regulates centrosome clustering via Stathmin, a Stat3 interactor and inhibitor of microtubule polymerization. Previous work has shown that Stat3 binds to Stathmin and inhibits the ability of Stathmin to depolymerize microtubules[36]. Since Stat3 inhibits Stathmin, depletion of Stathmin with siRNA should have a similar effect as Stat3 and therefore Stathmin siRNA should make Stattic less effective at inducing centrosome declustering. Stathmin knockdown with three out of four siRNAs increased centrosome clustering and Stattic was less effective in inhibiting centrosome clustering when Stathmin is knocked down (Fig. 3g,h). Stathmin siRNA sequences that were more effective at reducing Stathmin levels were also more effective at blocking Stattic effects on centrosome clustering. These results suggest that the Stat3–Stathmin pathway regulates centrosome clustering. However, it is not known whether Stat3 inhibitors block Stat3–Stathmin interaction. Stathmin co-immunoprecipitated Stat3, as shown previously[36]; however, this interaction was inhibited by Stattic (Fig. 3i).

Stathmin is widely considered as a protein that induces the depolymerization of microtubule polymers[37]. Since microtubules have been implicated in centrosome clustering[6], we tested whether Stattic treatment affected the ability of Stat3 to inhibit Stathmin depolymerase activity in vitro (Fig. 3j). Isolated tubulins spontaneously form long polymers in the presence of GTP and these polymers can be visualized by spotting them onto a microscope coverslip. Addition of recombinant Stathmin depolymerized these microtubule strands and Stat3 inhibited Stathmin depolymerase activity (Fig. 3j, upper), as has been previously shown[36]. We demonstrate that addition of Stattic or the clinically relevant Stat3 inhibitor BBI-608 both block the ability of Stat3 to inhibit Stathmin depolymerase function, allowing Stathmin to remain active to depolymerize microtubules (Fig. 3j, middle and bottom).

**Stat3–Stathmin centrosome clustering involves PLK1**. Since we demonstrated that Stattic indirectly promotes microtubule depolymerization, we next tested whether microtubule

**Figure 1 | Identification of KM08165 as a centrosome clustering inhibitor and chemical substructure analysis to identify Stattic.**
(**a**) Immunofluorescence images of cells treated with KM08165. Mitotic spindle morphology was observed by staining for pericentrin (green), α-tubulin (red) and DNA (Hoechst, blue). Scale bar, 8 µm. (**b**) Quantification of KM08165-dependent inhibition of centrosome clustering in cells derived from various origins including breast cancer (BT-549), myeloma (RPMI-8226, OPM-2) and freshly isolated normal primary human mammary epithelial cells (primary HMECs). n = 3 biological replicates, ≥128 cells per condition. Statistical significance was tested between untreated and KM08165-treated groups with analysis of variance (ANOVA). (**c**) Quantification of cell viability 3-(4,5-dimethylthiazol-2-yl)-2,5-diphenyltetrazolium bromide (MTT) assay in invasive mammary tumour cells (BT-549), normal mammary cells (MCF-10A) and primary human mammary epithelial cells (primary HMECs) treated with KM08165. n = 3 biological replicates. Statistical significance was tested between BT-549 and the other cell lines at different concentrations of KM08165 with ANOVA. (**d**) Chemical structures of KM08165 analogues and in silico determined chemical substructures. (**e**) Percentage of compound-treated BT-549 cells with declustered centrosomes. n = 3 biological replicates. Statistical significance was tested between untreated and compound-treated groups with ANOVA. (**f**) Relative viability of cells treated with KM08165 analogues and computed substructures. n = 3 biological replicates. Statistical significance was tested between untreated and compound-treated groups with ANOVA. (**g**) Images of mitotic cells treated with dimethylsulphoxide (DMSO; control), KM08165 and Stattic. Mitotic spindle morphology was observed by immunofluorescence staining for pericentrin (green), α-tubulin (red) and DNA (Hoechst, blue). Scale bar, 8 µm. *P < 0.05; **P < 0.01. Error bars represent s.d.

depolymerization is responsible for the ability of Stattic to induce centrosome declustering. To do this, we treated cells with 1 μM Stattic in the presence of the microtubule stabilizing drug paclitaxel (Fig. 4a). Paclitaxel alone induced a slight increase in the number of cells with declustered centrosomes, as has been shown previously[38]. However, paclitaxel could not rescue Stattic-induced centrosome declustering at any concentration tested, suggesting that Stat3–Stathmin function in centrosome clustering is largely independent of the role of Stathmin as a microtubule depolymerase.

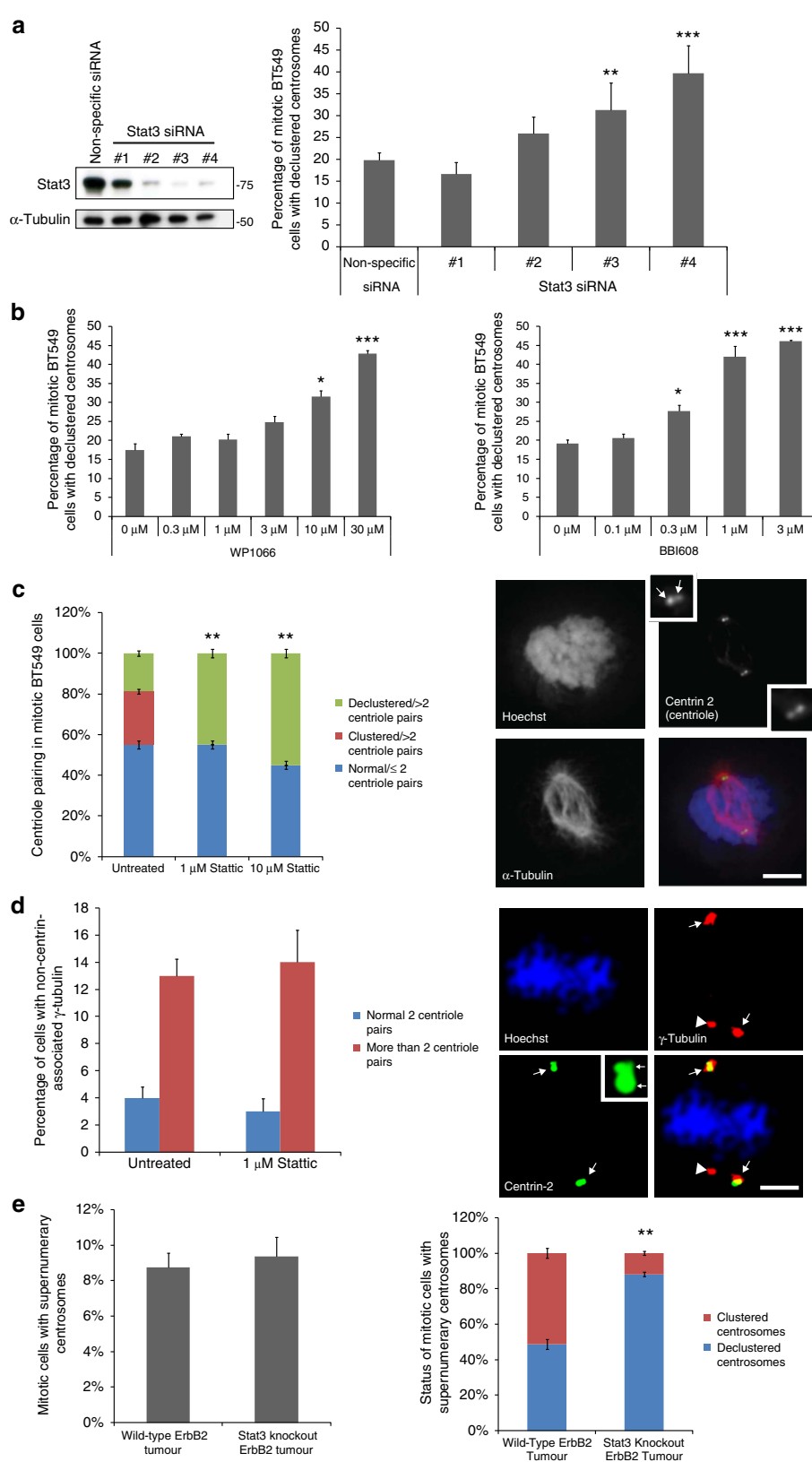

We had independently screened a library of 400 validated protein kinase inhibitors for their ability to inhibit centrosome clustering in mitotic BT-549 cells and implicated five protein kinases as potential mediators of centrosome clustering (Fig. 4b). We examined whether any of these hits could be linked to Stat3–Stathmin signalling. Of these hits, PLK1 and aurora kinase A (AURKA) have been previously implicated in mitotic centrosome positioning[39,40] and so were viewed as the most promising leads. If either of these kinases is in the Stat3–Stathmin pathway, Stattic should inhibit their activation. We therefore treated cells with a range of Stattic concentrations and measured ratios of phosphorylated (active) to total PLK1 and AURKA using phospho-specific antibodies and a fluorimetry-based assay (Fig. 4c). Stattic inhibited relative PLK1 phosphorylation by ∼40% but did not affect AURKA phosphorylation, suggesting that PLK1 is the potential downstream kinase in the Stat3–centrosome clustering pathway. In addition, Stat3 siRNA reduced relative phospho-PLK1 levels and constitutively active Stat3C but not kinase-dead Stat3-Y705F could rescue the effects of Stat3 siRNA on relative phospho-PLK1 levels (Fig. 4d). Stattic inhibition of phospho-PLK1 was confirmed by western blot analysis (Fig. 4e; Supplementary Fig. 4b) and immunofluorescence microscopy (Supplementary Fig. 4a).

Interestingly, Stathmin was previously demonstrated to inhibit PLK1 activity[41]. To confirm Stathmin inhibition of PLK1 activity and determine whether Stat3 is directly involved in Stathmin–PLK1 signalling, the mechanistic effects of Stattic, Stat3 and Stathmin on PLK1 activity were assessed using recombinant proteins and an *in vitro* ELISA-based assay for PLK1 activity (Fig. 4f; Supplementary Fig. 4c). Equimolar Stathmin had little effect on PLK1 activity but when the Stathmin concentration was doubled, PLK1 activity decreased by 44%. Since Stathmin levels are thought to be much higher than PLK1 levels in cells[41], this concentration of Stathmin was viewed as an acceptable level for subsequent testing. We next tested whether Stattic and Stat3 affected Stathmin-mediated inhibition of PLK1. When Stat3 was added at an equimolar concentration to Stathmin, PLK1 activity returned to near normal baseline activity levels. Furthermore, addition of Stattic blocked this Stat3-dependent effect (Fig. 4f, right).

PLK1 inhibition is known to effect mitotic spindle morphology[42], inducing either multipolar/declustered or monopolar spindles. We tested the PLK1 inhibitor BI-2536 at 10 and 100 nM concentrations and quantified mitotic spindle morphology in cells with the normal two centrosomes and with supernumerary centrosomes (Fig. 4g). BI-2536 (10 nM) was chosen as it is approximately the concentration where PLK1 activity is 50% inhibited[42] and Stattic treatment inhibited PLK1

by close to half (Fig. 4c). At 10 nM, BI-2536 inhibited centrosome clustering in cells with supernumerary centrosomes (Fig. 4g, right) while the spindle morphology in cells with two centrosomes was mostly normal (Fig. 4g, left). When the concentration of BI-2536 was increased to 100 nM, monopolar spindles became the predominant morphology, suggesting that partial and acute PLK1 inhibition cause distinct but overlapping multipolar or monopolar mitotic spindle morphologies. In support of this observation, we found that knockdown of PLK1 levels with three different siRNAs all induced primarily monopolar spindles but when the siRNA concentration used for transfection was diluted fivefold to reduce knockdown efficacy, declustered centrosome phenotypes were observed in cells with supernumerary centrosomes (Fig. 4g,h).

To further confirm that PLK1 is involved in Stat3-related centrosome clustering, we transiently overexpressed Myc-Tagged PLK1 in BT-549 cells that were also treated with 3 μM Stattic. Myc-PLK1-expressing cells were resistant to the effects of Stattic on centrosome clustering (Fig. 4i).

On the basis of the results so far, we have uncovered a supernumerary centrosome clustering pathway involving Stat3, Stathmin and PLK1 (Fig. 4j). PLK1 regulates a number of centrosomal proteins including γ-tubulin and therefore we examined the role of γ-tubulin in our pathway.

**Stat3–PLK1 acts on γ-tubulin to cluster centrosomes.** γ-Tubulin was imaged in mitotic cells with supernumerary centrosomes (Fig. 5a,b) and normal bipolar mitotic cells and the level of γ-tubulin at centrosome was quantified. As shown in previous research, cells with supernumerary centrosomes had lower γ-tubulin intensity per centrosome compared to regular bipolar cells (Fig. 5c), although the decrease we observed was smaller than a previous study[39]. Stattic reduced γ-tubulin levels in both normal bipolar mitotic cells as well as mitotic cells with supernumerary centrosomes. Since cells with two centrosomes cannot be declustered yet still had reduced γ-tubulin levels, centrosome declustering cannot be the cause of the γ-tubulin loss observed. Stat3 siRNA treatment of BT-549 cells reduced centrosomal γ-tubulin levels and stable expression of Stat3C, but not Stat3-Y705F, rescued the effects of Stat3 siRNA on γ-tubulin levels (Fig. 5c; Supplementary Fig. 5a). To examine PLK1 regulation of γ-tubulin, we measured γ-tubulin levels in BI-2536-treated cells (Fig. 5b,c). BI-2536 (10 nM) reduced centrosomal γ-tubulin levels to a similar level found with Stat3 inhibition. BI-2536 (100 nM) acutely reduced γ-tubulin levels. These results suggest that the Stat3–PLK1 pathway regulates γ-tubulin at the centrosome.

**Figure 2 | Stat3 is required for centrosome clustering.** (**a**) Western blot of Stat3 and tubulin (loading control) of BT-549 cells treated with Stat3 siRNAs (left). Quantification of the percentage of mitotic Stat3 siRNA-treated cells with declustered centrosomes (right; n = 3 biological replicates, ≥600 cells per condition). Statistical significance was tested between non-specific siRNA-treated and Stat3 siRNA-treated groups with analysis of variance (ANOVA). (**b**) Quantification of inhibition of centrosome clustering in cells treated with the Stat3 inhibitors WP1066 and BBI-608 (n = 5 biological replicates, ≥1,000 cells per data point). Statistical significance was tested between untreated and compound-treated groups with ANOVA. (**c**) Quantification of the number of centriole pairs in Stattic-treated cells (left; n = 4 biological replicates, 80 cells per condition). Statistical significance was tested between untreated and Stattic-treated groups with ANOVA. Immunofluorescence images of centriole pairs (centrin-2, green), tubulin (red) and DNA (Hoechst, blue) in a normal mitotic BT-549 cell (right). Inset: magnified image of centrin-2 stain. Arrows point to centrioles. Scale bar, 8 μm. (**d**) Quantification of the percentage of BT-549 cells with non-centrin-associated γ-tubulin (left panel) and immunofluorescence images (right panels) of centrin-2 (green), γ-tubulin (red) and DNA (Hoechst, blue). Scale bar, 5 μm. Arrows point to γ-tubulin with centrin-2 staining, arrowheads point to γ-tubulin staining without centrin-2 staining. Inset: magnified image of centrin-2 stain with arrows pointing to centrioles. n = 4 biological replicates, 100 cells per condition. Statistical significance was tested between untreated and Stattic-treated groups with Student's T-test. (**e**) Analysis of supernumerary centrosomes in Stat3 − / − ErbB2 induced mouse mammary tumours. The percentage of cells with supernumerary centrosomes was quantified as described in Experimental Procedures (left panel; n = 4 mice per condition, 80 cells per condition). In separate scoring, the percentage of supernumerary centrosome cells with clustered or declustered centrosomes was quantified (right; n = 4 mice per condition, 80 cells per condition). Statistical significance was tested between wild-type and Stat3-knockout groups with Student's T-test. *P < 0.05; **P < 0.01; ***P < 0.001. Error bars represent s.e.m.

To demonstrate that Stat3/PLK1 effects on spindle morphology are due to γ-tubulin, we treated BT-549 cells with γ-tubulin siRNAs (Fig. 5d,e) and examined centrosome morphology. γ-Tubulin siRNA-treated mitotic cells with the normal two centrosomes frequently formed monopolar spindles (Fig. 5e, left),

similar to the morphology seen in cells with acute PLK1 inhibition. γ-Tubulin siRNA-treated mitotic cells with supernumerary centrosomes had either monopolar or declustered centrosomes but very few clustered centrosomes (Fig. 5e, right). These results are similar to the effects we observed with acute

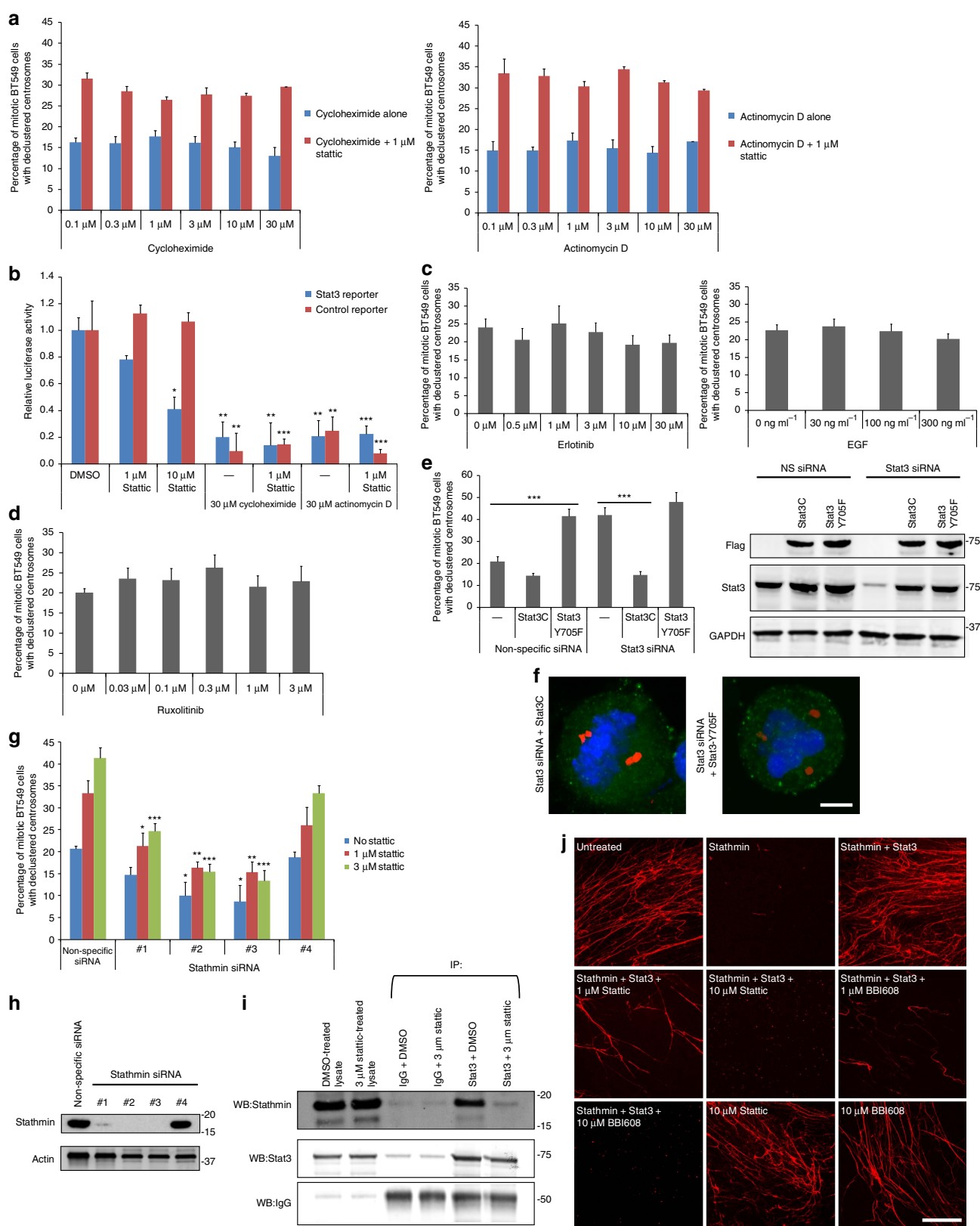

PLK1 inhibition (Fig. 5e compared to Fig. 4g), demonstrating that loss of γ-tubulin has similar effects on spindle morphology as loss of PLK1.

The primary function of γ-tubulin is to nucleate microtubules[43] and therefore a reduction in γ-tubulin by Stat3 inhibition should reduce microtubule density at the centrosome. Consistent with this, we observed a statistically significant reduction in the number of cells with long astral microtubules when BT-549 cells were treated with 3 μM Stattic (Fig. 5f,g).

PLK1 has been previously demonstrated to regulate centrosome positioning by maintaining γ-tubulin and astral microtubules[39]. We have confirmed these results and implicate this pathway in Stat3 signalling.

**Cells with amplified centrosomes are sensitive to Stat3.** We next explored whether Stat3 inhibition affects the viability of cancer cells with supernumerary centrosomes. To experimentally control centrosome amplification, we generated human breast cancer MDA-MB-231 cells that inducibly express PLK4 under a doxycycline-dependent promoter. PLK4 expression is a frequently used tool to induce cells to increase the number of centrosomes per cell[7,44] and we found that induced PLK4 expression increased the per cent of cells with supernumerary centrosomes from 37 to 94% (Fig. 6a), whereas a centrosome amplification-deficient PLK4 mutant[45] (PLK4[1−608]) had no effect on centrosome amplification (Supplementary Fig. 5b). PLK4[1−608] is a truncated but catalytically active version of PLK4 that does not dimerize and is therefore unable to induce centrosome amplification[45].

To confirm that centrosome clustering is indeed inhibited by Stattic, we scored centrosome clustering in Stattic-treated MDA-MB-231-PLK4 cells, with and without doxycycline-induced centrosome amplification. Doxycycline-induced MDA-MB-231-PLK4 had primarily clustered centrosomes, but Stattic treatment inhibited this centrosome clustering (Fig. 6b). We also examined whether Stattic-induced centrosome clustering inhibition persists throughout the cell cycle by counting the percentage of telophase cells that are multipolar (Supplementary Fig. 5c). Multipolar cell division is one of the ways that cells with declustered centrosomes can progress in mitosis[46]. We observed a statistically significant increase in the percentage of telophase cells that are multipolar, demonstrating that centrosome declustering persists in mitosis.

Using this doxycycline-inducible system, we determined cell viability in Stattic-treated MDA-MB-231-PLK4 cells with and without centrosome amplification using MTT assays (Fig. 6c,d;

Supplementary Fig. 5d,e). The MTT assay is a colorimetric measure of the metabolic conversion of the tetrazolium dye MTT into a purple formazan dye and is a common method for measuring cell viability since only living cells perform the enzymatic dye conversion. As has been shown previously[47,48], we found that treatment with high concentrations of Stattic or BBI-608 strongly affected cell viability, independent of centrosome amplification. In contrast, low level Stat3 or BBI-608 inhibition was significantly more effective at reducing cell viability in MDA-MB-231-PLK4 cells with centrosome amplification (PLK4 + Dox) versus cells without induced centrosome amplification (PLK4 no-Dox). Furthermore, uninduced and induced MDA-MB-231-PLK4[1−608] had equivalent cell viability with Stat3 inhibitors, demonstrating that centrosome amplification is specifically responsible for the change in cell viability.

## Discussion

In summary, we have elucidated a pathway involving Stat3, Stathmin and the mitotic kinase PLK1 that regulates centrosomal γ-tubulin levels which allows centrosomes to position themselves together[39] and form a clustered, bipolar spindle (summarized in Supplementary Fig. 4d). Stat3, by inhibiting Stathmin, activates the pathway that in turn relieves PLK1 inhibition, allowing PLK1 to increase γ-tubulin levels at the centrosome.

Previous studies show that PLK1 inhibition induces collapsed monopolar[49] or multipolar/declustered[50] spindle phenotypes. We demonstrate dose-dependent relationships between declustered and monopolar spindle phenotypes. Only supernumerary centrosome cells display declustered/multipolar spindles with partial PLK1 inhibition whereas all cells display monopolar spindle phenotypes when PLK1 is fully inhibited, independent of centrosome amplification. This demonstrates that PLK1 inhibition is particularly effective in cells with supernumerary centrosomes.

Stat3 is an essential factor in cancer initiation, growth and metastasis[17]. While Stat3 is primarily a transcription factor for cancer-promoting genes, Stat3 has other roles in the cytoskeleton, mitochondria and nucleus[36,51,52]. Here we demonstrate that centrosome clustering is a Stat3-dependent function, revealing a new mechanism to target cancer cells by Stat3 inhibitors.

The Stat3 inhibitor BBI-608 is currently in multiple phase III clinical trials[53,54]. Here we demonstrate that cancer cells with supernumerary centrosomes are inhibited to a significantly greater extent by BBI-608 than cancer cells without centrosome

**Figure 3 | Stat3 induces centrosome clustering via Stathmin and is independent of Stat3 transcription factor function.** (**a**) Quantification of centrosome declustering in BT-549 cells treated with the protein synthesis inhibitor cycloheximide (left panel; $n = 5$ biological replicates, $\geq 1,000$ cells per data point) and the transcription inhibitor actinomycin D (right panel; $n = 5$ biological replicates, $\geq 1,000$ cells per data point). (**b**) Transcriptional reporter activity of cells treated with Stattic, cycloheximide or actinomycin D. $n = 4$ biological replicates. Statistical significance was tested between untreated and compound-treated groups with analysis of variance (ANOVA). (**c**) Quantification of centrosome declustering in mitotic BT-549 cells treated with epidermal growth factor (EGF; left; $n = 5$ biological replicates, $\geq 1,000$ cells per data point) or the EGF receptor inhibitor erlotinib (right; $n = 5$ biological replicates, $\geq 1,000$ cells per data point). Statistical significance was tested between untreated and compound-treated groups with ANOVA. (**d**) Quantification of centrosome declustering in BT-549 cells treated with the Jax1/2 inhibitor ruxolitinib. $n = 5$ biological replicates, $\geq 1,000$ cells per data point. Statistical significance was tested between untreated and ruxolitinib-treated groups with ANOVA. (**e**) Left: quantification of centrosome declustering in Stat3 siRNA-treated mitotic BT-549 cells stably expressing Flag-Stat3C or Flag-Stat3-Y705F ($n = 5$ biological replicates, 250 cells per condition). Right: western blot of Flag, Stat3 and GAPDH (loading control) using lysates from Stat3 siRNA-treated cells stably expressing Flag-Stat3C or Flag-Stat3-Y705F. Statistical significance between the indicated groups was tested using ANOVA. (**f**) Immunofluorescence images of Stat3 siRNA-treated BT-549 cells stably expressing Flag-Stat3C (left) or Flag-Stat3-Y705F (right). Pericentrin (red), Flag (green) and DNA (Hoechst, blue). Scale bar, 4 μm. (**g**) Quantification of centrosome declustering in BT-549 cells treated with Stathmin siRNA ± Stattic. $n = 4$ biological replicates, $\geq 800$ cells per condition. Statistical significance was tested between untreated and Stattic-treated groups with ANOVA. (**h**) Western blot of Stathmin and actin (loading control) using lysates from Stathmin siRNA-treated cells. (**i**) Immunoprecipitation of IgG (control) or Stat3 from BT-549 cells using endogenous protein and western blotted for Stathmin, Stat3 and IgG. (**j**) *In vitro* tubulin polymerization assay using purified proteins. Microtubules were grown in the presence of the microtubule depolymerase Stathmin (HIS-tagged), Stat3 (GST-tagged), Stattic and BBI-608. HIS-Stathmin and GST-Stat3 were used at 1.8 and 0.36 μM concentrations, respectively. Scale bar, 8 μm. *$P < 0.05$; **$P < 0.01$; ***$P < 0.001$. Error bars represent s.e.m.

amplification. Since centrosome amplification is frequently observed in tumour biopsies from cancer patients[4], it will be interesting to examine whether patients with tumours containing centrosome amplification respond better to BBI-608 and other Stat3 inhibitors in clinical trials.

## Methods

**Cell culture.** All cell lines were from American Type Culture Collection except the MCF10-A (Sigma Aldrich, St Louis, MO, USA), OPM-2 (DSMZ, Braunschweig, Germany), the normal primary human bone marrow cells (Stem Cell Technologies, Vancouver, Canada) and the primary HMECs (see below). None of the cell lines used are listed in the Database of Cross-contaminated or Misidentified Cell Lines[55].

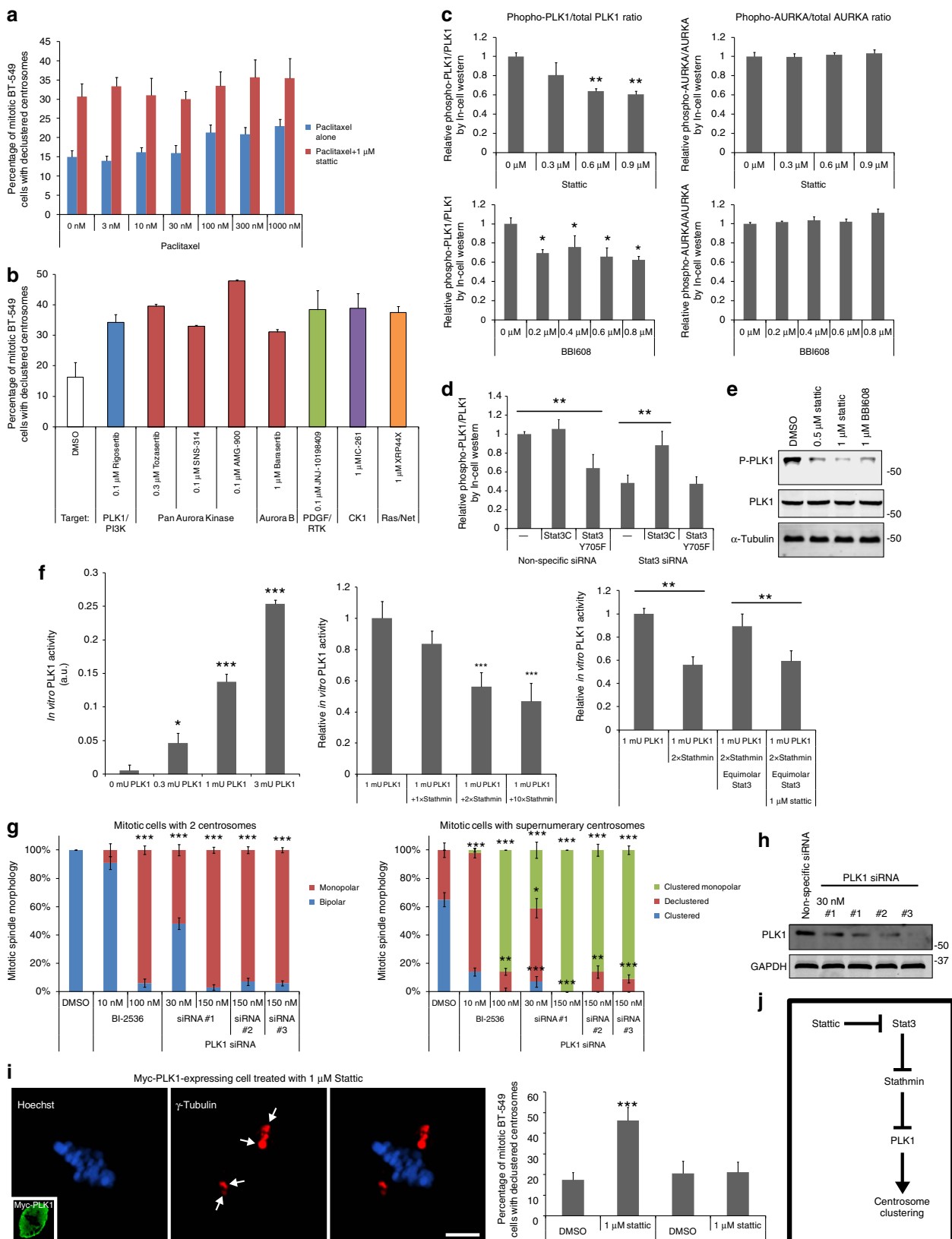

The LookOut Mycoplasma PCR Detection Kit (Sigma Aldrich) was used to check for mycoplasma contamination in the cell lines used. Samples of the cell lines used in this paper were sent to a commercial testing facility (Genetica, Burlington, NC, USA) to ensure that the cell lines used were authentic. This facility used short tandem repeat DNA fingerprinting to determine cell line identity.

For the primary normal HMECs, histologically normal discard breast tissue from women who had undergone reduction mammoplasty surgery was collected with informed consent and approval of the University of British Columbia Ethics Board. Tissues were enzymatically dissociated overnight to generate mammary organoids. Mammary single cells were isolated from organoids following rapid enzymatic and mechanical dissociation. Proliferative epithelial cells were enriched using a 3-day preculture method in SF7 media containing DMEM/F12 supplemented with 10 ng ml$^{-1}$ epidermal growth factor, 10 ng ml$^{-1}$ cholera toxin, 1 μg ml$^{-1}$ insulin and 0.5 μg ml$^{-1}$ hydrocortisone described previously[56]. Cells were trypsinised and viable HMECs isolated by fluorescence-activated cell sorting were cultured in SF7 media supplemented with 5% fetal bovine serum, overnight, prior to compound treatment[16,57].

Non-HMECs were grown in either DMEM, RPMI or McCoy's 5A media containing 10% fetal bovine serum as recommended by American Type Culture Collection except MCF-10A, which were grown according to a published protocol[58].

**Metabolic conversion of KM08165 to Stattic.** Metabolic conversion of KM08165 into Stattic was determined using human liver microsomes (0.5 mg ml$^{-1}$). To begin the reaction, human liver microsomes were mixed with 10 μM KM08165 and the metabolic reaction was started with NAPDH. Loss of substrate was used as a measure of metabolic conversion of KM08165. KM08165 and metabolically converted Stattic were detected using UPLC-UV.

**DNA and siRNA constructs and viral transduction.** MYC-PLK1 was a gift from Erich Nigg[59] (Addgene, Cambridge, MA, USA). Myc-PLK1 was transfected into BT-549 cells with Lipofectamine 2000 (ThermoFisher, Waltham, MA, USA). siRNAs were all from Qiagen (Valencia, CA, USA) and transfected into cells using siLentfect (Bio-Rad, Hercules, CA, USA) as described previously[60]. To score centrosome clustering in these MYC-PLK1 cells, mouse anti-γ-tubulin (Sigma Aldrich) and rabbit anti-Myc (Cell Signaling Technology) antibodies were used. γ-Tubulin has been frequently used to score centrosome clustering in previous papers[38,61–63].

pLenti-CMV-TetR Blast was a gift from Eric Campeau[64] (Addgene). pLenti-CMV/TO-Neo-PLK4 and pLenti-CMV/TO-Neo-PLK4$^{1−608}$ were gifts from David Pellman[7]. Lentiviral Flag-Stat3C and Flag-Stat3-Y705F were gifts from Linzhao Cheng[65] (Addgene). To generate doxycycline-inducible MDA-MB-231-PLK4 cells, we infected cells first with pLenti-CMV-TetR-Blast lentiviral particles and selected cells using 10 μg ml$^{-1}$ blasticidin to generate stable cell lines that express the Tet repressor. We next infected these TetR-expressing cells with pLenti-CMV/TO-Neo-PLK4 lentiviral particles and selected with 800 μg ml$^{-1}$ G418 to create a stable cell line that inducibly expresses PLK4. The bulk population was selected and no clones were chosen, as per previous methods[7]. Lentiviral constructs were packaged using 293-T cells by co-transfection with psPAX2 and pMD2.G (gifts from Didier Trono (Addgene)) using TransIT-LTI (Mirus, Madison, WI, USA) and following the Addgene recommended protocol.

**Phenotypic screen for centrosome clustering inhibitors.** The Maybridge chemical library was used for phenotypic screening of centrosome clustering and the OICR kinase inhibitor library (Selleck Chemicals, Houston, TX, USA) was used for screening kinase inhibitors that potentially effect centrosome clustering.

A total of 7000 BT-549 cells were plated onto black-sided 96-well plates (BD Falcon, San Diego, CA, USA) overnight then treated with the compound library using a robotic pinning instrument. After 5 h incubation, cells were fixed in PBS with 4% paraformaldehyde and 0.01% Triton-X100, and stained using Hoechst 33342 (for locating cells), mouse TG3 antibody (mitotic marker) and rabbit anti-pericentrin (centrosome marker; Abcam ab4448, 1:200). Pericentrin has been extensively used in previous studies for scoring centrosome amplification and clustering[66–69], and it was the most reliable marker for centrosomes out of all the antibodies we tried. Alexa-568 anti-mouse and Alexa-488 anti-rabbit (ThermoFisher) were used for secondary antibodies. Processed immunofluorescent cells on 96-wells were then imaged using a Cellomics Array Scan VTI automated imaging microscope (ThermoFisher). Fifteen fields were captured for each well using a × 10 objective.

To count the number of mitotic cells with declustered centrosomes, the Thermo Scientific Compartmental Analysis program was configured and used as described previously[16]. All cells in the images taken were automatically analysed which was typically 2,000–5,000 cells.

**MTT assay and clonogenic and colony-forming unit assays.** For the MTT cell viability assay, 1,000–3,000 cells (or 5,000 primary HMECs) were cultured on 96-well plates overnight and compounds were added for 72 h except for the MDA-MB-231-PLK4 cells where Stattic or BBI-608 was added for 48 h. Cell viability was assessed by treating cells with 5 mg ml$^{-1}$ thiazolyl blue tetrazolium bromide for 4 h then solubilization solution (10% SDS and 0.01 N HCl) overnight. After the MTT assay was finished, absorbance at 570 and 660 nm wavelengths was measured with a spectrophotometer. Absorbance at 660 was subtracted from absorbance at 570 to remove background signal and samples were normalized relative to untreated cells (without dimethylsulphoxide).

The clonogenic and colony-forming unit assays were conducted as described previously[16]. More than 50 cells were counted for each concentration of KM08165 in each cell type. Cells were plated in six-well plates (50 cells per well) and let adhere to the plastic for several hours. KM08165 concentrations were added in duplicates. After 9 days, cells were fixed and stained with CrystalViolet. For non-adherent cancer cell lines, cells were plated in methylcellulose based media and treated with KM08165 in duplicates (500 or 1,000 cells per plate). After 14 days of incubation, colonies consisting of > 50 cells were counted under a microscope. Colony-forming unit assays were performed according to the manufacturer's instruction using primary normal human bone marrow cells (CD34-enriched) and cultured for 14 days in MethoCult H4435 (Stem Cell Technologies) in duplicates (500 or 1,000 cells per plate).

**PLK1 activity assay.** Recombinant GST-Stat3 protein was from Abnova (Taipei City, Taiwan), recombinant HIS-Stathmin protein was from Prospec (Burnaby, BC, Canada) and active recombinant PLK1 was from SignalChem. PLK1 Assay/Inhibitor Screening Kit (Cyclex, Nagano, Japan) was used following manufacturer's recommendations and the colorimetric assay in 96-well format was read on a SpectraMax i3 plate reader (Molecular Devices, Sunnyvale, CA, USA). A concentration of 1 mU PLK1 corresponds to an 8.9 nM concentration and therefore 1 × Stathmin corresponds to 8.9 nM concentration, 2 × Stathmin corresponds to 17.8 nM concentration and 10 × Stathmin corresponds to 89 nM concentration. Equimolar Stat3 was 17.8 nM concentration, corresponding to the 2 × Stathmin concentration used in the assay (Fig. 4f).

**In-Cell Western fluorometric assay.** The In-Cell Western assay measures scanned fluorescence intensity of paraformaldehyde-fixed cells in a 96-well plate format using infrared dye-labelled antibodies. For the In-Cell Western, BT-549

---

**Figure 4 | Stat3/Stathmin act through PLK1 to regulate centrosome clustering.** (**a**) Quantification of centrosome clustering in paclitaxel-treated BT-549 cells with or without Stattic ($n = 6$ biological replicates, ≥1,200 cells per data point). (**b**) Positive kinase inhibitor 'hits' identified from a screen of 400 known kinase inhibitors. $n = 2$ biological replicates, ≥400 cells per data point. Error bars (s.d.). PLK1 and aurora kinase A (AURKA) were selected as potential downstream effector candidates of Stat3–Stathmin. (**c**) Quantification of phosphorylated to total PLK1 and AURKA ratios in Stat3-inhibitor-treated BT-549 cells using In-Cell Western assays ($n = 4$ biological replicates). (**d**) Quantification of phospho-PLK1 to PLK1 ratios in BT-549 cells stably expressing Stat3C or Stat3-Y705F and treated with Stat3 siRNA, as determined using In-Cell Western assays ($n = 8$ biological replicates). (**e**) Western blot of phospho-PLK1 and PLK1 from lysates of BT-549 cells treated with Stattic or BBI-608. Tubulin is a loading control. (**f**) ELISA-based quantification of in vitro PLK1 activity. Quantification of PLK1 activity assay using increasing concentrations of PLK1 (left panel; $n = 4$ biological replicates). Quantification of relative PLK1 activity using 1 mU PLK1 and increasing amounts of Stathmin (middle panel; $n = 4$ biological replicates). Quantification of recombinant PLK1 activity in the presence of recombinant Stathmin, Stat3 and Stattic (right panel; $n = 4$ biological replicates). 1 mU (milli unit) = 1 nmole of phosphate incorporated min$^{-1}$ mg$^{-1}$. (**g**) Quantification of the effects of low and high concentrations of the PLK1 inhibitor BI-2536 and PLK1 siRNA on centrosome clustering in mitotic BT-549 cells ($n = 4$ biological replicates, 80 cells per condition). (**h**) Western blot of PLK1 siRNA-treated BT-549 cells stained for PLK1 and GAPDH (loading control). (**i**) Immunofluorescence images of γ-tubulin (red), DNA (Hoechst, blue) and Myc-Tag (inset, green) in a mitotic BT-549 cell transiently overexpressing Myc-tagged PLK1 and treated with 1 μM Stattic (left panels). Arrows point to clustered supernumerary centrosomes. Scale bar, 5 μm. Quantification of the percentage of Myc-PLK1-expressing mitotic BT-549 cells with declustered centrosomes (right panel; $n = 4$ biological replicates, 120 cells per condition). (**j**) Diagram of potential Stat3–Stathmin–PLK1 pathway. *$P < 0.05$; **$P < 0.01$; ***$P < 0.001$. Error bars represent s.e.m. Statistical significance was tested between the indicated groups with analysis of variance in all graphs.

cells were grown in 96-well plates and treated with inhibitors for 5 h before being fixed with 4% paraformaldehyde then ice-cold methanol and stained with the indicated primary antibodies. Secondary detection was with IRDye 800CW anti-Mouse and IRDye 680 anti-Rabbit (Licor, Lincoln, NE, USA). Plates were then scanned using the Licor Odyssey Imaging system and the ratio of phosphorylated to total protein was determined.

**Stat3 reporter assay.** Stat3 transcriptional activity was determined by transfecting a Stat3 transcriptional reporter, pTATA TK-Luc (gift from Jim Darnell (Addgene)), and a non-specific reporter, pRL-TK (Promega, Madison, WI, USA), together using Lipofectamine 2000. Cells were left to recover for 5 h then inhibitors were added for 5 h. Transcriptional activity was then assayed using the Dual Luciferase Reporter Assay (Promega) according to the manufacturer's instructions

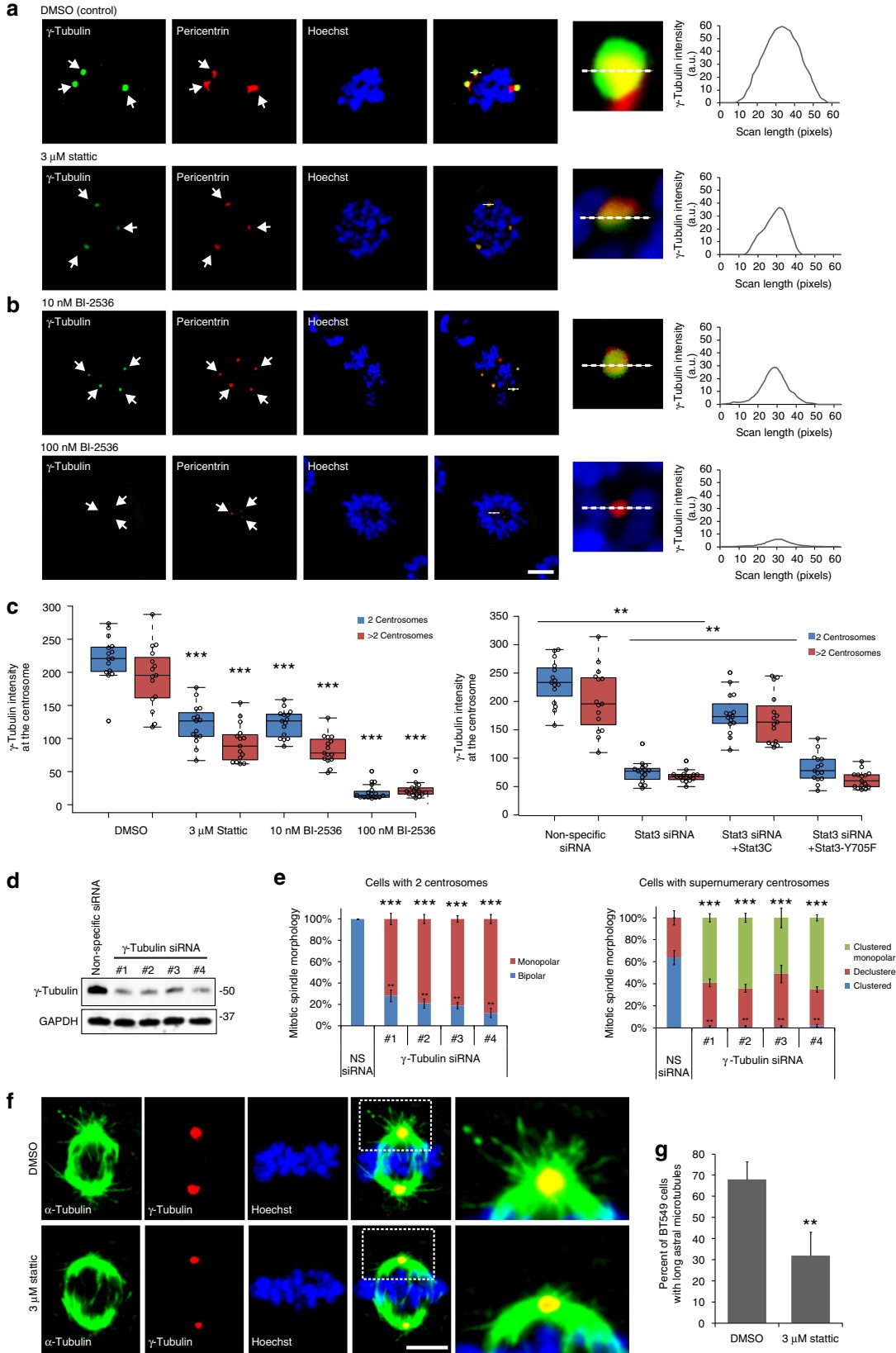

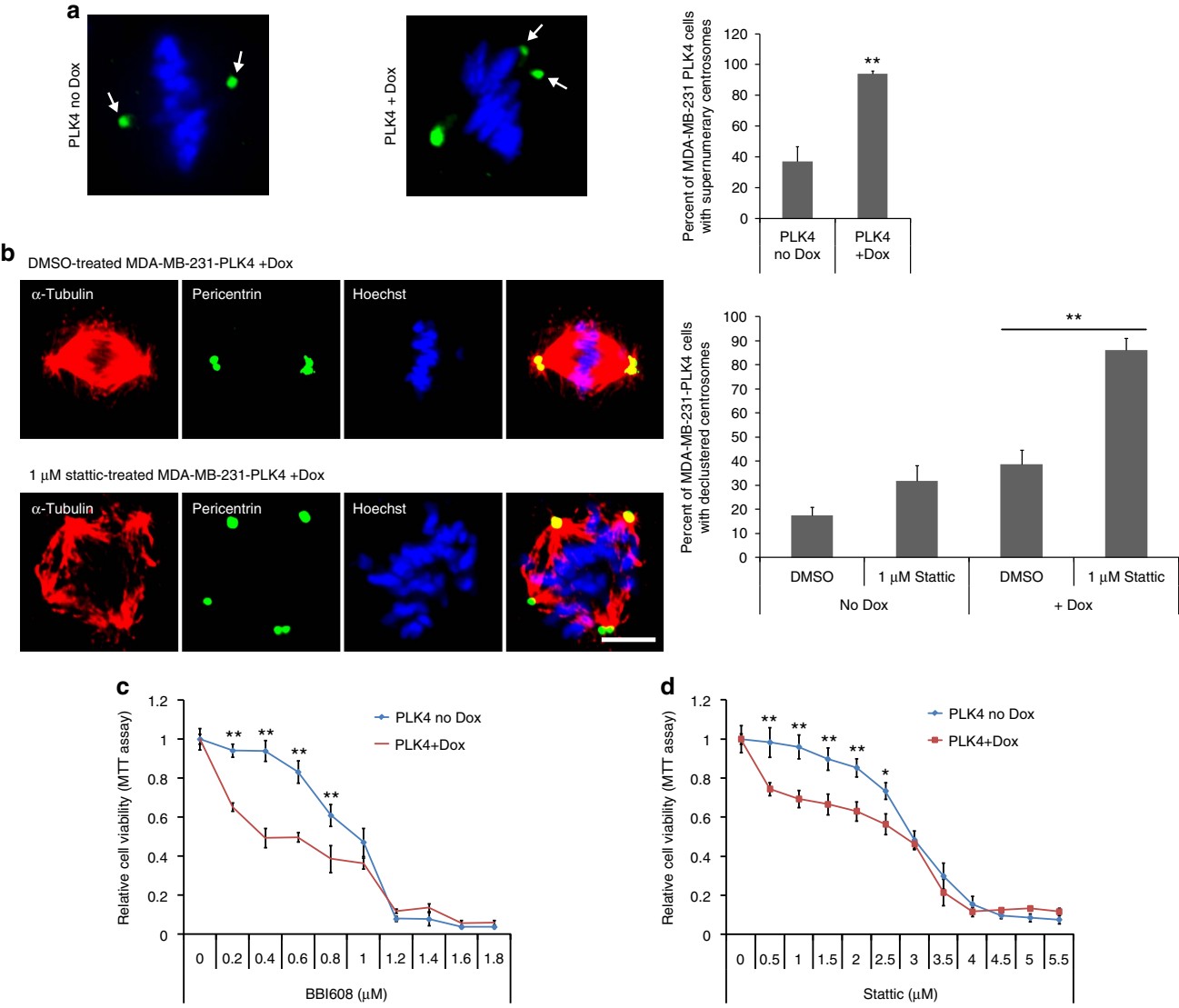

**Figure 6 | Cells with centrosome amplification are more sensitive to Stat3 inhibitors.** (**a**) Left: representative immunofluorescence images of MDA-MB-231 cells with doxycycline-induced, PLK4-dependent centrosome amplification. Pericentrin (green) and DNA (Hoechst, blue). Scale bar, 3 μm. Right: quantification of the number of doxycycline-induced (Dox) MDA-MB-231-PLK4 cells with supernumerary centrosomes. $n = 5$ biological replicates, ≥100 cells per condition. (**b**) Left: representative immunofluorescence images of dimethylsulphoxide (DMSO)-treated and Stattic-treated MDA-MB-231 cells with doxycycline-induced, PLK4-dependent centrosome amplification. Tubulin (red), pericentrin (green) and DNA (Hoechst, blue). Scale bar, 3 μm. Right: quantification of the number of Stattic-treated doxycycline-induced (Dox) MDA-MB-231-PLK4 cells with declustered centrosomes. $n = 4$ biological replicates, 160 cells per condition. (**c**) Quantification of cell viability (MTT assay) in BBI-608-treated MDA-MB-231-PLK4 cells with (PLK4 + Dox) or without (PLK4 no-Dox) doxycycline-induced PLK4-608 expression. $n = 6$ biological replicates. Statistical significance was tested between the no-Dox and Dox groups with analysis of variance (ANOVA). (**d**) Quantification of cell viability (MTT assay) in Stattic-treated MDA-MB-231-PLK4 cells with (PLK4 + Dox) or without (PLK4 no-Dox) doxycycline-induced centrosome amplification. $n = 6$ biological replicates. Statistical significance was tested between no-Dox and Dox groups with ANOVA. $*P < 0.05$; $**P < 0.01$. Error bars represent s.e.m.

**Figure 5 | The Stat3–PLK1 pathway regulates centrosomal γ-tubulin and astral microtubules.** Immunofluorescence images of pericentrin (red), γ-tubulin (green) and Hoechst (blue) in mitotic supernumerary centrosome BT-549 cells treated with Stattic (**a**) or BI-2536 (**b**). Graphs of line scans (right panels), dashed white lines denote the scanned area, arrows point to centrosomes. Scale bar, 3 μm. (**c**) Quantification of fluorescence intensities of γ-tubulin in mitotic cells treated with Stattic or BI-2536 (top) or Stat3 siRNA and Stat3C or Stat3-Y705F (bottom). Data points (circles). Centre lines (medians). Box limits indicate the 25th and 75th percentiles. Whiskers extend 1.5 times the interquartile range from the 25th and 75th percentiles. $n = 15$ biological replicates. Statistical significance was tested between control and treated groups with analysis of variance (ANOVA). (**d**) Western blot of γ-tubulin siRNA-treated BT-549 cells stained for γ-tubulin and GAPDH (loading control). (**e**) Quantification of the mitotic spindle morphologies of γ-tubulin siRNA-treated mitotic BT-549 cells with two centrosomes (left) or supernumerary centrosomes (right). $n = 5$ biological replicates, 100 cells per condition. Statistical significance was tested between non-specific and γ-tubulin siRNA-treated groups with ANOVA. (**f**) Immunofluorescence images of α-tubulin (green), γ-tubulin (red) and DNA (Hoechst, blue) in mitotic BT-549 cells with normal centrosome number that have been treated with dimethylsulphoxide (DMSO; top) or Stattic (bottom). Magnified areas show astral microtubules. Scale bar, 3 μm. (**g**) Quantification of the percentage of normal (two centrosome) mitotic BT-549 cells with long astral microtubules. $*P < 0.05$; $**P < 0.01$; $***P < 0.001$. Error bars represent s.e.m.

except the amount of cell lysate used was increased 20-fold. Cells transfected for 5 h without inhibitor treatment were used as controls for background subtraction.

**In vitro microtubule polymerization.** Purified calf tubulin and rhodamine-labelled tubulin (Cytoskeleton, Denver, CO, USA) were combined at a 10:1 ratio and added to tubulin assembly buffer (80 mM Pipes (pH 6.8), 0.5 mM EGTA, 2 mM MgCl2 and 10% glycerol) with 1 mM GTP for 30 min at 37 °C to generate microtubules in vitro. Recombinant GST-Stat3, HIS-Stathmin, Stattic and BBI-608 were added before the microtubules were assembled. A 1:5 ratio of GST-Stathmin (0.36 μM) to HIS-Stathmin (1.8 μM) was used following a published method[36]. After 30 min incubation, microtubules were fixed using 0.5% glutaraldehyde and the microtubules were spotted onto glass coverslips precoated with mounting media and imaged.

**Inhibitors and immunofluorescence microscopy.** Stattic (Sigma Aldrich), KM08165 (Maybridge, Trevillet, Cornwall, UK), KM08176 (Maybridge), KM08138 (Maybridge), thionapthene (Maybridge), WP1066 (Millipore, Billerica, MA, USA), 5,15-DPP (Millipore), BBI-608 (Abcam, Cambridge, UK), Erlotinib (Cayman Chemical, Ann Arbor, MI, USA), Ruxolitinib (Cayman Chemical), BI-2536 (Selleck Chemicals), GF15 (Millipore), hydroxyurea (Sigma Aldrich), cycloheximide (Cayman Chemical), actinomycin D (Sigma Aldrich), piceatannol (Cayman Chemical) and paclitaxel (Sigma Aldrich) and were all dissolved in dimethylsulphoxide at recommended concentrations.

For immunofluorescence microscopy, cells were fixed in Dithiobis(succinimidyl propionate), then 4% paraformaldehyde[60] or were fixed in −20 °C methanol for 15 min and rehydrated in PBS. For phospho-PLK1 staining, cells were fixed in 4% paraformaldehyde, then −20 °C methanol (both for 15 min). Primary antibodies for immunofluorescence microscopy were: rabbit anti-pericentrin (Abcam ab4448; 1:2,000), mouse anti-α-tubulin (Sigma Aldrich T9026; 1:1,000), mouse anti-γ-tubulin (Abcam ab11316; 1:200), mouse anti-PLK1 (pT210; BD Bioscience 558400; 1:1,000), rabbit anti-PLK1 (Novus Biologicals NB100-56651; 1:500), rabbit anti-Aurora A (pT288; Abcam ab18318; 1:500), mouse anti-Aurora A (Sigma Aldrich A1231; 1:200), mouse anti-Centrin-2 (Santa Cruz Biotechnology sc-293192; 1:100), rabbit anti-centrin-2 (Santa Cruz Biotechnology sc-27793; 1:100) and rabbit anti-Myc (Cell Signaling Technology 2276S; 1:500). DNA was stained using Hoechst 33342 (ThermoFisher). Secondary antibodies were from Jackson ImmunoResearch or ThermoFisher. Preparation of mouse TG3 antibody from hybridoma culture supernatant was described previously[70]. Immunofluorescence images were obtained using Nikon Eclipse TI or Zeiss LSM780 confocal microscopes or a Zeiss Colibri epifluorescence microscope and processed with Image J. Microscopic analysis of centrosome declustering was conducted as described previously[16].

**γ-Tubulin and astral microtubule measurement.** Levels of γ-tubulin at the centrosome were measured by focusing on the middle (maximal size and intensity) of an individual centrosome, recording an image, then in Image J, placing a 1 μm circle at the centroid of the centrosome and measuring integrated pixel intensity. Background signal was subtracted and was measured by placing a circle in an area of the cell away from the centrosomes and measuring γ-tubulin integrated pixel intensity. Cells where individual centrosomes could not be distinguished were excluded from analysis.

To plot γ-tubulin intensity, BoxPlotR was used (http://boxplot.tyerslab.com). Note that line scans of the type shown in Fig. 5a,b were not used for quantifying γ-tubulin intensity at the centrosome.

BT-549 cells were treated for 4 h with Stattic then fixed, stained and scored for the presence of long astral microtubules in mitotic cells. To ensure consistent results and to accurately determine which microtubules were astral microtubules, only cells with two centrosomes were scored and supernumerary centrosome cells were ignored for this assay.

**Statistics.** Statistical analysis was performed using Microsoft Excel and GraphPad Prism 6. Student's t-test was used to determine P values for all data involving comparisons between two groups. The t-tests were for unequal variance and were two-tailed. Two-way analysis of variance with post hoc test was used to compare multiple groups (as noted in figure legends). Sample variance was estimated using error bars in graphs. Sample sizes for cell and animal data were chosen based on previous literature in the relevant field.

**Immunoprecipitation and western blot.** For western blotting, mouse anti-Stat3 (Novus Biologicals; 1:1,000), mouse anti-Stathmin (Santa Cruz Biotechnology sc-55531; 1:200), rabbit anti-Stathmin (Novus Biologicals NB110-57602; 1:1,000), mouse anti-α-tubulin (Sigma Aldrich; 1:2,000), rabbit anti-GAPDH (Sigma Aldrich; 1:500), mouse anti-PLK1 (pT210; BD Bioscience; 1:1,000), rabbit anti-PLK1 (Novus Biologicals; 1:500), rabbit anti-Beta-Actin (Sigma Aldrich; 1:5,000) and mouse anti-γ-tubulin (Abcam; 1:1,000) were used. Cells were lysed using 1% NP-40 in Tris-buffered saline. SDS–polyacrylamide gel electrophoresis and western blotting were performed as described previously[60]. Immunoprecipitations were performed using 1% NP-40 in Tris-buffered saline with added phosphatase inhibitors (sodium vanadate and sodium fluoride) and protease inhibitor

(Roche, Basel, Switzerland) and 5 μg mouse anti-Stat3 antibody (Santa Cruz Biotechnology). Uncropped scans of western blots are shown in Supplementary Fig. 6.

Liver-derived microsome assays and UPLC-UV detection of KM08165 conversion into Stattic was performed by the Centre for Drug Research and Development (Vancouver). In silico prediction of KM08165 substructures was performed with the help of the Centre for Drug Research and Development.

**Mammary-specific Stat3 knockout/ErbB2-overexpressing tumours.** The transgenic mouse breast cancer model using ErbB2 to induce mammary-specific tumours was described previously[22]. Mice were housed at the Royal Victoria Hospital animal care facility and all experiments were conducted in accordance with the animal care guidelines at the Animal Resource Centre of McGill University. Stat3 conditional mice harbouring LoxP-flanked Stat3 were crossed with transgenic NIC strain mice to generate mice harbouring mammary epithelial cells with activated ErbB2 that simultaneously express Cre recombinase resulting in ErbB2-overexpressing Stat3-knockout mouse tumours (Stat3flx/flx/NIC). Tumours were detected with biweekly palpation and animals were killed 6 weeks after initial detection. Stat3flx/flx/NIC tumour growth and metastasis has been previously characterized[22].

Tumours from mice were fixed in 10% neutral buffered formalin, embedded in paraffin wax and sectioned at 4 μm. Antigen retrieval was performed with citrate buffer and using a microwave. Sections were blocked in PBS with 5% goat serum and 0.5% casein. For staining tumour sections, Hoechst 33342, anti-mouse pericentrin (BD Biosciences; 1:100) and rabbit anti-Phospho-Histone H3 (Ser10; Cell Signalling 9701; 1:200) were used. Cells with abnormally large centrosomes and cells with high background pericentrin stain were excluded from analysis.

**Data availability.** All other remaining data are available within the article and Supplementary Files, or available from the authors upon request.

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

## Acknowledgements

This work is supported by grants to S.D. from the CIHR and CCSRI, with funds raised by the Canadian Cancer Society and the Centre for Drug Research and Development (CDRD). We thank Michael Cox for help with confocal microscopy and Iain Cheeseman and David Pellman for cells and reagents (see Extended Experimental Procedures). We thank Connie Eaves for providing mammoplasty tissues for fresh isolation of HMECs. Mammoplasty tissue was obtained with the assistance of D. Wilkinson and surgeons J. Sproul, P. Lennox, N. Van Laeken and R. Warren. We thank the CDRD for providing the protein kinase inhibitor library.

## Author contributions

E.J.M. and J.A.G. performed the experiments except E.K. performed experiments in Fig. 1, Fig. 4b and Supplementary Fig. 1. E.J.M., E.K. and S.D. designed the experiments. E.J.M. and S.D. wrote the manuscript. A.B. and M.R. supervised the compound library screens and provided the compound libraries. N.K. isolated primary HMECs and contributed to Fig. 1b,c. W.J.M. supervised the Stat3-knockout tumours and contributed histological samples.

## Additional information

**Competing interests:** The authors declare no competing financial interests.

