## [Peer Review File · Nature Communications]

Reviewers' comments:

Reviewer #1 (Remarks to the Author):

Centrosome amplification is a common feature of human tumours and has been recently associated with tumour incidence and progression in mouse models. Importantly, to survive, cancer cells need to efficiently cluster their extra centrosomes, thus opening new opportunities to the development of cancer therapies.

In this manuscript, the authors describe a novel role for stat3 in the regulation of centrosomes clustering. They found that a small molecule against stat3 prevents efficient clustering in cancer cell lines and mouse tumour models. Stat3 inhibition prevents centrosome clustering via stathmin and PLK1 regulation. Because stat3 inhibitors are in clinical trial, understanding how their efficacy could be improved, for example by stratifying patients according to centrosome amplification, could have important consequences to the clinic.

This manuscript makes important contributions to the field. However, there are several experimental issues and unclear conclusions, particularly concerning the mouse data. Also, the mechanism identified by the authors is not clear. For example the role of γ -tubulin in this process is not well demonstrated. But perhaps the major weakness of this manuscript is the fact that the authors do not convincingly show that de-clustering of extra centrosomes is what is killing the cells treated with stattic, which is the major conclusion of this manuscript. In my opinion, there are several important issues the authors should address prior to publication (see specific comments below).

Major comments:

#1. The authors initially describe the identification of the KM08165 as a centrosome de-clustering drug, which is then converted into the stat3 inhibitor, Stattic. However, the data suggesting that KM08165 is in fact converted into Stattic is not shown. Can the authors show this data (which is the starting point of this study)?

#2. The centrosome de-clustering is quantified in prometa/metaphase cells by the authors. However, by doing so, the authors cannot exclude that the cells with de-clustered centrosomes in metaphase are not able to cluster the centrosomes prior to anaphase onset. This is important because centrosome de-clustering in metaphase might only indicate a delay in centrosome clustering. To demonstrate that Stattic prevents clustering, the authors need to do live cell imaging at least in one of the more used cell lines to quantify the percentage of cells that undergo multipolar or bipolar mitosis.

#3. The tumour data in this manuscript is not strong. For example, in Figure 2d, the authors used a tumour model (ErbB2) to quantify centrosome clustering where only 8% of cells that extra centrosomes. Thus, in each 80 mitotic cells that the authors are quantifying in the tumours, there are only approximately 6 cells with extra centrosomes per condition? These low numbers of cells cannot be used to assess accurately centrosome clustering. Either the authors chose a different model where centrosome amplification is visible or as it is these data should be removed.

#4. It has been previously demonstrated that preventing efficiently centrosome clustering kills cancer cells in a way that it is proportional to their percentage of centrosome amplification. Thus, the concentration of Stattic the authors should use should only reduce viability proportionally to the number of cells that contain extra centrosomes. What is exactly the % of centrosome amplification (with centrin staining) of BT-549, RPMI-8226 and OPM-2? BT-549 cells should not have more than 50% of centrosome amplification, so if the drug is killing more than 50% of cells (see graph 1c) at 10uM then clearly the drug is doing something more than de-clustering.

#5. The role of stat3 in regulating stathmin is an interesting finding. The data on figure 3i that shows that stathmin and stat3 interact and that this interaction can be inhibited by stattic is incomplete. It is missing the control of the input lysates before the IPs. This is essential to see whether stattic is preventing stathmin-stat3 interaction or actually is reducing stathmin levels in

the lysates. It is also unclear why the authors keep using different concentrations of the drug in all experiments.

#6. The authors claim that PLK1 is important for the stat3-dependent clustering because overexpressing myc-PLK1 prevents de-clustering by stattic (figure 4i). However, the only thing the authors offer to support their claim is an IF image of what seems to be a normal metaphase spindle, without any quantitation. The authors need to quantify the % of centrosome clustering in cells expressing myc-PLK1 and stained for centrin, so that cluster can be assessed.

#7. The authors propose that decreased γ -tub at the centrosomes in cells treated with stattic prevents centrosome clustering. However, the data presented does not support this conclusion. First, the authors never compare γ -tub staining in cells also stained for centrin to assess centrosome amplification (figure 5a-c). Pericentrin is not a good marker for centrosome number. Also, it is possible that, because γ -tub is not abundant in the cytoplasm, if centrosomes are de-clustered less γ -tub is associated per centrosomes since it can become diluted. Thus, less γ -tub per centrosome in cells with de-clustered centrosomes could be a consequence of de-clustering and not a cause. In addition, siRNA of γ -tub (figure 5e) leads mostly to monopolar spindles and the % of de-clustered centrosomes in control cells and cells depleted of γ -tub does not change, so it is uncertain why the authors say it induces de-clustering? It is also unclear why a moderate decrease in γ -tub (judge by the IF image in figure 5f) in cells treated with stattic should abolish astral microtubules? This needs to be quantified as well as the IF should include EB1 staining to assess astral microtubules more reliably.

#8. It is confusing why the authors chose to use a cell line (MDA-231) that has been reported by several people to have around 40-50% of cells with extra centrosomes to overexpress PLK4 (figure 5g-h). It has been shown that induction of centrosome de-clustering in this cell line can lead to 40-50% of loss of cell viability. Therefore, the expectation would be that concentrations of stattic that kill 40% of MDA-231 cells should kill close to 100% of MDA-231 cells that overexpress PLK4. This is not what the authors observe in figure 5h, and the cell death data shown cannot be explained by the de-clustering of extra centrosomes.

#9. The data presented in figure 5i is very unclear. First, why the authors are showing the data for 50k cells since this does not lead to tumours? Secondly, for the 500k experiments, is there any evidence of cell viability after treatment with the drug for 7 days? Are the authors injecting dead cells in the animals? The relevance of this experiment is also uncertain since usually people inject cells and then treat the animal with the drug, to mimic cancer treatment in patients. As it is, this data does not add to their findings.

Minor comments:

#10. The stats used for all the graphs in this manuscript, with the exception of 2d, 5g and 5b, where the t-test is correct, are incorrect. T-test are inaccurate to use due to family wise error in most of the graphs presented here. Instead, two-way ANOVA with post-hoc test should be used. Plus, it should be clear in the graphs which conditions are being compared.

#11. Why the authors represent the graphs with centrosome amplification with lines between the different points? This is only possible if the x-axis is an actual scale. These graphs (1E, 1F, 2b, 3a, 3c, 3d, 4a, 5h, S1e, S1f, S2b, S2c, S5c, S5d, S5e) should be changed to columns.

#12. Page 2: In the first line "In many types of cancers... patient outcomes" they should also include the reference: Chan JH, "A Clinical Overview of Centrosome Amplification in Human Cancers" *Int J Biol Sci.* 2011; 7(8): 1122-1144. 2011

#13. Page 2: Work from David Glover lab (published on Open Biology) showing that inducible PLK4 overexpression accelerates tumourigenesis in mice should be added (where Sercin et al was added).

#14. Page 3: In the first sentence the authors speculate on centrosome declustering as being an attractive therapeutic strategy, but they have no reference. There are numerous papers that have shown that impairing centrosome clustering leads to cell death in cells with supernumerary centrosomes. References should be added here.

#15. Page 5: The last sentence in the second paragraph states that the authors observed cases of centrosome fragmentation – however, they do not have evidence that this is due to fragmentation. It could be due to induction of centrosome amplification instead.

#16. Page 12-13: The description of PLK4 needs to be clarified. Overexpression of PLK4 is used as a tool for centrosome amplification, as PLK4 regulates of centrosome duplication. There is also no description of what the PLK41-608 is?

Reviewer #2 (Remarks to the Author):

Considering that many tumor cells harbor excess centrosomes and yet manage to form bipolar spindles by centrosome clustering, the induction of centrosome declustering (which then leads to formation of multipolar spindles and cell death) has emerged as a potentially attractive therapeutic strategy. In this manuscript, Morris, Dedhar and coworkers report a hitherto unexpected role of Stat3 in controlling the extent of centrosome clustering in (tumor) cells displaying excess centrosomes. Their work originates with a screen aimed at identifying small molecule inhibitors of centrosome clustering. From an analysis of substructures of the primary hit compound (KM08165), they zoom in on Stattic, a metabolic derivative of KM08165, that has previously been characterized as an inhibitor of Stat3 (Schust et al. op. cit). Following up on this observation, the authors then search for a pathway through which Stat3 might enhance centrosome clustering. Surprisingly, they find that this novel function of Stat3 does not depend on canonical upstream signaling (i.e. EGF-induced activation of EGFR or Jak2 kinase), although it does depend on Stat3 dimerisation, nor does it depend on regulation of transcription. They then follow up on a previous report suggesting that Stat3 inhibits the ability of Stathmin to depolymerize microtubules. Having provided evidence that Stattic indeed affects this functionality of Stathmin in vitro, they however claim that the Stat3-Stathmin function in centrosome clustering is largely independent of microtubule depolymerization in vivo. Then, the story takes a somewhat unexpected twist. In an attempt to find downstream effectors of Stat3-Stathmin, the authors survey some 400 protein kinase inhibitors for a possible role in the regulation of centrosome clustering, and this leads them to focus on Plk1. They show that Stattic inhibits Plk1 activity (as measured through analysis of activating Plk1 phosphorylation) and that Plk1 affects centrosome clustering through its (previously established) regulation of gamma-tubulin recruitment. Finally, the authors engineer human breast cancer cells (MDA-MB-231) allowing them to induce centrosome amplification through inducible expression of Plk4, the master regulator of centriole duplication. Using this system they show that the presence of multiple centrosomes sensitizes cells (as well as tumors in NOD/SCID mice) to low doses of Stat3 inhibitors. In conclusion, the authors describe a novel pathway important for the regulation of centrosome clustering. This pathway links Stat3 to Stathmin and Stathmin to Plk1. Their data lead them to propose that Stat3 inhibitors (which are currently in phase III trials) might be particularly effective on tumors showing centrosome amplification.

In my opinion, the paper is very well written: by linking their data to pre-existing "bits and pieces" from the literature, the authors present a "story" that strikes me as intriguing. Overall, I find the data to be of high quality and the interpretation of the results mostly plausible, albeit perhaps not always 100% conclusive. My specific criticism concerns the following points:

Major points:

1. I have a major issue with the results described in Figure 2c/text on page 5. The authors correctly state that a change in the number of separated "centrosomes" per cell could be due to centrosome declustering, but also to centrosome "amplification" or "fragmentation". They then use centrin staining to count centriole pairs and use their results to argue against "fragmentation". The problem with this interpretation is that centrin staining measures centriole numbers and thus constitute a proxy for centrosome amplification – a priori, it says nothing about fragmentation of pericentriolar material. The fact that centriole numbers do not change thus argues against overduplication, not against fragmentation! One common assay for centrosome fragmentation is staining for gamma-tubulin. In fact, fragmentation is typically defined as an increase in gamma-

tubulin-positive foci that are able to nucleate microtubules but lack centrioles (and hence lack centrin staining) – because fragmentation only affects ‘pericentriolar material’.

2. A second major problem concerns the claim that regulation of centrosome clustering by Stat3 is independent of transcription (Figure 3a,b) and text on page 6. This is a key conclusion of the paper, but I am afraid I am unable to evaluate the validity of this conclusion. In particular, I could not find a clear description of how these experiments were done. A priori I would expect that inhibition of transcription or translation should severely impair cell cycle progression and so I wonder how the authors are able to observe mitoses in apparently undiminished frequency? When describing the experiments aimed at confirming the efficacy of the inhibitors, they report that inhibitors were applied for several hours. Was the same protocol used prior to analysis of mitotic cells? It might help to provide a schematic that illustrates the temporal sequence (and duration) of inhibitor treatments, relative to the time of analysis. For their “story” to stand, it is absolutely critical that the authors exclude a transcriptional effect of Stat3 inhibition on centrosome clustering/declustering.

3. Related to the above point, a schematic illustrating the experimental setup used to explore the role of canonical EGF and Jak2 signaling upstream of Stat3 would also be helpful. While I do not consider myself an expert in the Stat3 field, I was under the impression that Stat3 dimerization is controlled by tyrosine kinase receptor activation and downstream Jak2 signaling. If this were correct (but I may be wrong...), how do the authors reconcile the importance of Stat3 dimerization for centrosome clustering with the independence of this Stat3 function from upstream signaling/Jak2?

4. In Figure 3J, the authors confirm previous literature showing that, in vitro, Stat3 inhibits Stathmin-induced microtubule depolymerization, and that Stat3 inhibitors block this function. To explore the existence of a similar mechanism in vivo, they then ask whether Paclitaxel influences Stathmin-induced centrosome declustering, and because this microtubule stabilizing drug does not produce much of an effect, they conclude that the Stat3-Stathmin function in centrosome clustering is largely independent of microtubule depolymerization. I find this argument rather weak. I would expect that the Stathmin literature should offer a more direct way to explore the impact of Stathmin/Stat3/Stathmin on the stability of microtubule polymers in cells (e.g. using differential extraction)?

5. Figure 4 presents several histograms plotting ratios of Western blot signals. Specifically, the authors compare signals obtained with phospho-specific and pan-kinase antibodies, and they show one example of these signals in Figure 4e. Although I recognize that antibodies recognizing specific “activating” phosphorylation sites in kinases can represent valuable tools, these assays are not the most direct or rigorous ways for measuring kinase activities. Thus, while the histograms in Figure 4 collectively suggest a “clear case”, I think the authors should also provide the corresponding primary data (i.e. the corresponding Western blots) in a supplementary Figure.

6. Figure 4i requires quantification. From this single exemplary IF panel it is impossible to conclude that Myc-Plk1 expressing cells are resistant to the effects of Stathmin on centrosome clustering.

Minor points:

1. IF panels of Figure 5: reading the corresponding text, I would have expected to see a comparison of gamma-tubulin levels in mitotic cells that spontaneously display 2 centrosomes or more than 2 centrosomes (not just cells having undergone drug treatment).

2. One comment about “spinning the tale”: having argued against microtubule depolymerization being downstream of Stat3-stathmin, the authors turn to explore a role of kinases – as if this was the most logical next step. However, when reading this passage, their argumentation struck me as less than compelling. My suspicion is that the authors had independently performed kinase inhibitor screens, pointing to Plk1 and Aurora, and later saw an opportunity to link the

corresponding data to their work on Stat3-stathmin. If this were the case, there would be absolutely nothing wrong with this – all good papers present data in a most logical order, not necessarily reflecting history. Nevertheless, I suspect that some re-wording (perhaps a more transparent presentation of the reasons for shifting emphasis to kinases) might smoothen this particular transition in the paper.

3. It might be helpful to briefly “define” MTT assays in the main text.

4. p. 14: the authors state “This demonstrates that Plk1 inhibition is “uniquely” effective in cells with supernumerary centrosomes” – I think what they mean is that “Plk1 inhibition is “particularly” effective... (“Uniquely” seems the wrong word; after all, Plk1 inhibition is also effective in cells with 2 centrosomes – only less so).

Reviewer #3 (Remarks to the Author):

The use of the “medicinal chemistry” is not correct, because the derivatives were purchased (page 3).

The structure of the nitro group is not NO₂H, but NO₂. This applies to a total of 4 molecules in Figure 1d and Supplementary Figure 1b.

The term “thionaphthene” for the derivative lacking the nitro group is incorrect.

The authors use STAT3C as a constitutively active STAT3 mutant and assign this activity to constitutive dimerization. However, PMID 16956893 reports that STAT3C is not a covalent dimer, but instead has elevated activity due to higher DNA binding affinity. The authors might want to comment on this.

Reviewers' comments

Reviewer #1 (Remarks to the Author):

Centrosome amplification is a common feature of human tumours and has been recently associated with tumour incidence and progression in mouse models. Importantly, to survive, cancer cells need to efficiently cluster their extra centrosomes, thus opening new opportunities to the development of cancer therapies.

In this manuscript, the authors describe a novel role for stat3 in the regulation of centrosomes clustering. They found that a small molecule against stat3 prevents efficient clustering in cancer cell lines and mouse tumour models. Stat3 inhibition prevents centrosome clustering via stathmin and PLK1 regulation. Because stat3 inhibitors are in clinical trial, understanding how their efficacy could be improved, for example by stratifying patients according to centrosome amplification, could have important consequences to the clinic.

This manuscript makes important contributions to the field. However, there are several experimental issues and unclear conclusions, particularly concerning the mouse data. Also, the mechanism identified by the authors is not clear. For example the role of γ -tubulin in this process is not well demonstrated. But perhaps the major weakness of this manuscript is the fact that the authors do not convincingly show that de-clustering of extra centrosomes is what is killing the cells treated with stattic, which is the major conclusion of this manuscript. In my opinion, there are several important issues the authors should address prior to publication (see specific comments below).

Major comments:

Reviewer Comment:

#1. The authors initially describe the identification of the KM08165 as a centrosome de-clustering drug, which is then converted into the stat3 inhibitor, Stattic. However, the data suggesting that KM08165 is in fact converted into Stattic is not shown. Can the authors show this data (which is the starting point of this study)?

Author Response:

A table of this data has been inserted as Supplementary Figure 1e. Please see figure legend and Methods subsection "Metabolic conversion of KM08165 to Stattic" (page 1 of Methods Section) for an explanation of the assay performed.

Reviewer Comment:

#2. The centrosome de-clustering is quantified in prometa/metaphase cells by the authors. However, by doing so, the authors cannot exclude that the cells with de-clustered centrosomes in metaphase are not able to cluster the centrosomes prior to anaphase onset.

This is important because centrosome de-clustering in metaphase might only indicate a delay in centrosome clustering. To demonstrate that Stattic prevents clustering, the authors need to do live cell imaging at least in one of the more used cell lines to quantify the percentage of cells that undergo multipolar or bipolar mitosis.

Author Response:

It has been previously established that cancer cells with supernumerary centrosomes are able to continue dividing when centrosomes are declustered, either through tripolar mitoses^{1,2} or by reclustering later in mitosis through elongation of the cell in anaphase and telophase³. These processes are less efficient than regular bipolar mitosis and a range of defects are observed which lead to increased rates of cell death as well as slower growth in cells with declustered centrosomes³.

We are not claiming that all declustered centrosome cells die, just that the growth of cells with supernumerary centrosomes is more sensitive to Stat3 inhibitors relative to normal cells. We measured cell survival in Stattic-treated cells with and without induced centrosome amplification (Figure 6c and Supplementary Figure 5d). There was a statistically significant decrease in cell viability in Stattic treated cells with centrosome amplification but clearly the cells did not just all die and many were able to continue dividing after centrosome declustering. We note that a somewhat similar change in viability (and not complete cell death) was previously found with an established centrosome clustering inhibitor, GF-15⁴, using a similar assay. This suggests that centrosome clustering inhibition would be expected to cause growth inhibition and not cell death.

Reviewer Comment:

#3. The tumour data in this manuscript is not strong. For example, in Figure 2d, the authors used a tumour model (ErbB2) to quantify centrosome clustering where only 8% of cells that extra centrosomes. Thus, in each 80 mitotic cells that the authors are quantifying in the tumours, there are only approximately 6 cells with extra centrosomes per condition? These low numbers of cells cannot be used to assess accurately centrosome clustering. Either the authors chose a different model where centrosome amplification is visible or as it is these data should be removed.

Author Response:

We appreciate the reviewer's concern however we believe that clarifying how the data was collected should address this concern.

We first went through and counted the number of mitotic cells with normal or supernumerary centrosomes. 80 mitotic cells were counted from WT tumours and 80 mitotic cells were counted from Stat3 knockout tumours (Figure 2e, left panel). In both cases, these 80 mitotic cells were scored from a total of 4 tumours.

In a separate analysis, we went back and scored whether mitotic cells with amplified centrosomes had clustered or declustered centrosomes (Figure 2e, right panel). 80 mitotic cells with supernumerary centrosomes were counted for WT mice and 80 mitotic cells with supernumerary centrosomes were counted from Stat3-knockout mice. In both cases, tissue sections from 4 tumours were used.

The reason that there were two separate counts was because scoring normal versus amplified centrosomes is very straightforward whereas scoring clustered versus declustered is more complicated since the relative separation of the individual centrosomes needs to be assessed and this scoring has to be very consistent in all the tissue sections examined. Overall, we stained 10 tissue sections per tumour.

We have now added more detail to the figure legend for Figure 2e describing this experiment to clarify that the scoring for clustering was conducted independently of the scoring for supernumerary centrosomes.

Reviewer Comment:

#4. It has been previously demonstrated that preventing efficiently centrosome clustering kills cancer cells in a way that it is proportional to their percentage of centrosome amplification. Thus, the concentration of Stattic the authors should use should only reduce viability proportionally to the number of cells that contain extra centrosomes. What is exactly the % of centrosome amplification (with centrin staining) of BT-549, RPMI-8226 and OPM-2?

BT-549 cells should not have more than 50% of centrosome amplification, so if the drug is killing more than 50% of cells (see graph 1c) at 10uM then clearly the drug is doing something more than de-clustering.

Author Response:

The cell lines (BT-549, RPMI-8226, OPM-2) were chosen because they have been previously shown to be highly sensitive to centrosome clustering inhibitors⁴ and the multiple myeloma cell lines RPMI-8226 and OPM-2 have a high centrosome index⁵ that makes them sensitive to centrosome clustering inhibitors. We have updated the text on page 4 to discuss the reason why we chose these cell lines.

Stat3 has been shown to have many roles in the cell. It is most often characterized as a transcription factor that activates growth promoting or anti-apoptotic genes⁶. Many papers have shown that Stat3 inhibition induces cell death but the concentrations used are typically 5-10uM or greater. We have identified an additional way that Stat3 inhibition reduces cell viability: by interfering with centrosome

clustering in mitotic cells with supernumerary centrosomes. The declustering effect we observed occurs at a much lower concentration of Stattic versus Stattic concentrations that kill all cells in general. Based on the extensive Stat3 literature, we assume that 10uM Stattic is inhibiting cell viability by inhibiting other Stat3 related functions that are independent of centrosome amplification.

Thus while our data shows that Stat3 inhibitors induce centrosome declustering and reduce cell viability, we are not claiming that the effect on cell viability is entirely dependent on centrosome declustering. We now discuss the established roles of Stat3 in cell growth and survival in the text on page 3.

Reviewer Comment:

#5. The role of stat3 in regulating stathmin is an interesting finding. The data on figure 3i that shows that stathmin and stat3 interact and that this interaction can be inhibited by stattic is incomplete. It is missing the control of the input lysates before the IPs. This is essential to see whether stattic is preventing stathmin-stat3 interaction or actually is reducing stathmin levels in the lysates. It is also unclear why the authors keep using different concentrations of the drug in all experiments.

Author Response:

We have repeated the Stat3-Stathmin immunoprecipitation and have included lysates used for the immunoprecipitation (new Figure 3i). There was no change in Stathmin level when cells were treated with Stattic.

Reviewer Comment:

#6. The authors claim that PLK1 is important for the stat3-dependent clustering because overexpressing myc-PLK1 prevents de-clustering by stattic (figure 4i). However, the only thing the authors offer to support their claim is an IF image of what seems to be a normal metaphase spindle, without any quantitation. The authors need to quantify the % of centrosome clustering in cells expressing myc-PLK1 and stained for centrin, so that cluster can be assessed.

Author Response:

We have repeated the experiment on a larger scale and have now quantified the number of Stattic-treated, Myc-PLK1 expressing cells with declustered centrosomes. Myc-PLK1 was expressed in BT549 cells that were treated with 1uM Stattic. This is shown in the new Figure 4i.

Centrosome clustering was quantified using Gamma-Tubulin since we found it was a reliable marker of centrosomes in BT-549 cells and many researchers have previously used it to measure centrosome clustering⁷⁻¹⁰. We have added a comment to the Methods section that states Gamma-Tubulin has been extensively used to score centrosome clustering (Methods Section pages 1 and 2).

Reviewer Comment:

#7. The authors propose that decreased γ -tub at the centrosomes in cells treated with stattic prevents centrosome clustering. However, the data presented does not support this conclusion. First, the authors never compare γ -tub staining in cells also stained for centrin to assess centrosome amplification (figure 5a-c).

Author Response:

We have now performed this quantification as shown in the new Figure 2d (this was also performed in response to comments from Reviewer #2). We did not observe a change in the number of cells with non-Centrin associated Gamma-Tubulin when Stattic was used. We have also now added a sentence in the text on page 5 to discuss these new experiments showing that Stattic does not induce centrosome fragmentation.

Reviewer Comment:

Pericentrin is not a good marker for centrosome number.

Author Response:

Pericentrin has been extensively used in previous studies for scoring centrosome amplification and clustering¹¹⁻¹⁶ and it was the most reliable marker for centrosomes out of all the antibodies we tried. We have now cited references that used Pericentrin to score centrosomes clustering and amplification in the Method section (page 2 of the Methods section) to indicate that Pericentrin is an extensively used marker for centrosome clustering.

Reviewer Comment:

Also, it is possible that, because γ -tub is not abundant in the cytoplasm, if centrosomes are de-clustered less γ -tub is associated per centrosomes since it can become diluted. Thus, less γ -tub per centrosome in cells with de-clustered centrosomes could be a consequence of de-clustering and not a cause.

Author Response:

We also observed reduced Gamma-Tubulin in Stattic-treated cells with only 2 centrosomes (Figure 5c). Since cells with 2 centrosomes are not declustered, Gamma-Tubulin dilution due to declustering cannot be the sole cause of the observed loss of Gamma-Tubulin staining. Gamma-Tubulin levels were slightly reduced in cells with multiple centrosomes compared to normal cells which we interpret as

Gamma-Tubulin being diluted by the presence of multiple centrosomes. Dilution of Gamma-Tubulin due to centrosome amplification has been observed previously¹⁷ and was expected to occur.

We now discuss this point in the manuscript on page 12.

Reviewer Comment:

In addition, siRNA of γ -tub (figure 5e) leads mostly to monopolar spindles and the % of de-clustered centrosomes in control cells and cells depleted of γ -tub does not change, so it is uncertain why the authors say it induces de-clustering?

Author Response:

We are suggesting that Gamma-Tubulin siRNA had a similar effect on mitotic cells as acute PLK1 inhibition. Acute PLK1 inhibition (Figure 4g, right panel) caused almost entirely collapsed monopolar spindles with some declustered centrosome cells as well. Similarly, Gamma-Tubulin siRNA caused mostly collapsed monopolar spindles with some declustered centrosome cells as well (see Figure 5e, right panel). From this, we suggest that the effects of Gamma-Tubulin inhibition on spindle morphology were similar to PLK1 inhibition. Other data in the paper was used to suggest that there is a continuum between the declustered spindle morphology (which we relate to intermediate PLK1 inhibition and intermediate Gamma-Tubulin levels at the centrosome) and the collapsed monopolar spindle morphology (which we relate to acute PLK1 inhibition and very low levels of Gamma-Tubulin at the centrosome).

We have changed the text on page 13 to clarify this point.

Reviewer Comment:

It is also unclear why a moderate decrease in γ -tub (judge by the IF image in figure 5f) in cells treated with stattic should abolish astral microtubules? This needs to be quantified as well as the IF should include EB1 staining to assess astral microtubules more reliably.

Author Response:

We have now quantified the astral microtubule data using a published protocol¹⁸. Using this method, we found that a significantly lower percentage of Stattic-treated cells have long astral microtubules relative to DMSO (control) treated cell. This is shown in the new Figure 5g.

Previous papers have shown that reductions in PLK1 activity cause a reduction in Gamma-Tubulin and a reduction in microtubule number¹⁷. PLK1 acts on many centrosome-associated proteins¹⁹ and there

are likely multiple events involved in declustering. We were using Gamma-Tubulin and reduced astral microtubules as a known pathway that is affected.

This point is now discussed on pages 13 and 14 of the text.

Reviewer Comment:

#8. It is confusing why the authors chose to use a cell line (MDA-231) that has been reported by several people to have around 40-50% of cells with extra centrosomes to overexpress PLK4 (figure 5g-h). It has been shown that induction of centrosome de-clustering in this cell line can lead to 40-50% of loss of cell viability. Therefore, the expectation would be that concentrations of Stattic that kill 40% of MDA-231 cells should kill close to 100% of MDA-231 cells that overexpress PLK4. This is not what the authors observe in figure 5h, and the cell death data shown cannot be explained by the de-clustering of extra centrosomes.

Author Response:

Stattic is a Stat3 inhibitor. Stat3 inhibition has been shown by many papers to cause cell death by inhibiting expression of anti-apoptotic genes and to cause growth inhibition by interfering with the expression of growth promoting genes. The growth promoting and anti-apoptotic role of Stat3 is now discussed on page 3 of the text. We expected and observed Stat3-dependent inhibition of cell viability that was independent of centrosome amplification. It just happens that the Stattic-dependent inhibition of centrosome clustering occurs at a lower concentration versus the other effects Stattic has on the cell.

What we have shown is that there is a relative decrease in the viability of Stattic-treated MDA-MB-231 cells when centrosomes are amplified by PLK4 expression. This relative difference in viability is only seen at low concentrations of Stattic. At high concentrations of Stattic, all cells are affected (as expected). This is discussed on pages 14 and 15.

Reviewer Comment:

#9. The data presented in figure 5i is very unclear. First, why the authors are showing the data for 50k cells since this does not lead to tumours? Secondly, for the 500k experiments, is there any evidence of cell viability after treatment with the drug for 7 days? Are the authors injecting dead cells in the animals? The relevance of this experiment is also uncertain since usually people inject cells and then treat the animal with the drug, to mimic cancer treatment in patients. As it is, this data does not add to their findings.

Author Response:

To address this concern, we injected uninduced and induced MDA-MB-231-PLK4 cells into mice and treated the mice with different doses of Stattic. However, none of the Stattic dosing regimens had any effect on tumour growth, regardless of whether centrosomes were amplified or not. This was surprising because Stat3 has been shown to have a strong effect on cell growth and viability. Therefore, we think that the lack of effect of Stattic in vivo was likely due to the pharmacokinetics of the Stattic compound itself. While it will be important to establish the pharmacokinetics of Stattic treatment in our mouse models and using our cell lines, it is beyond the scope of this paper and will require a whole different study to establish.

Since we had already observed a differential response of uninduced and induced MDA-MB-231-PLK4 cells to Stattic in vitro (see Figure 6c), we decided to pre-treat cells with Stattic then inject them into mice and record the growth of the tumours that formed. This is shown in the new Figure 6d and Supplementary Figure 5e. Pre-treatment of MDA-MB-231-PLK4 cells with Stattic caused a decrease in tumour volume when comparing uninduced and induced cells, similar to the difference we observed in cells growing in vitro (Figure 6c). We assume that the induced Stattic-treated cells experienced disrupted growth due to defects that had been initiated during aberrant mitoses due to centrosome declustering, as has been shown in previous papers^{2,3}.

To ensure that injected cells were equally viable, cells were checked for viability as they were being counted and prepared for injection into mice. Trypan blue dye exclusion was used to assess cell viability. This is standard procedure for any animal experiment and is also standard procedure for routine cell counting. We used the number of live cells as our count for injecting into mice. We have updated the text of the Methods section (page 6) to describe the Trypan Blue stain for viability.

Minor comments:

Reviewer Comment:

#10. The stats used for all the graphs in this manuscript, with the exception of 2d, 5g and 5b, where the t-test is correct, are incorrect. T-test are inaccurate to use due to family wise error in most of the graphs presented here. Instead, two-way ANOVA with post-hoc test should be used. Plus, it should be clear in the graphs which conditions are being compared.

Author Response:

We have now indicated in the figure legends what is being compared in the graphs. We have now used two-way ANOVA to generate significance values for the relevant graphs. This has been updated in the manuscript and in the Methods section (page 5 of Methods section).

Reviewer Comment:

#11. Why the authors represent the graphs with centrosome amplification with lines between the different points? This is only possible if the x-axis is an actual scale. These graphs (1E, 1F, 2b, 3a, 3c, 3d, 4a, 5h, S1e, S1f, S2b, S2c, S5c, S5d, S5e) should be changed to columns.

Author Response:

The scale used is an approximation of a logarithmic scale (1, 3, 10, 30, 100, 300 etc) and is commonly used in dose response studies. Line graphs have been used in conjunction with a scale of this type in many papers²⁰⁻²⁴. We have now changed the line graphs to column style graphs when this type of scale was used.

Reviewer Comment:

#12. Page 2: In the first line “In many types of cancers... patient outcomes” they should also include the reference: Chan JH, “A Clinical Overview of Centrosome Amplification in Human Cancers” Int J Biol Sci. 2011; 7(8): 1122–1144. 2011

Author Response:

We have inserted the additional reference into the manuscript.

Reviewer Comment:

#13. Page 2: Work from David Glover lab (published on Open Biology) showing that inducible PLK4 overexpression accelerates tumourigenesis in mice should be added (where Sercin et al was added).

Author Response:

We have inserted the additional reference into the manuscript.

Reviewer Comment:

#14. Page 3: In the first sentence the authors speculate on centrosome declustering as being an attractive therapeutic strategy, but they have no reference. There are numerous papers that have shown that impairing centrosome clustering leads to cell death in cells with supernumerary centrosomes. References should be added here.

Author Response:

We have inserted additional references into the manuscript.

Reviewer Comment:

#15. Page 5: The last sentence in the second paragraph states that the authors observed cases of centrosome fragmentation – however, they do not have evidence that this is due to fragmentation. It could be due to induction of centrosome amplification instead.

Author Response:

Using a published centrosome amplification assay, we did not see any effect of Stattic on centrosome amplification (Supplementary Figure 2e and f). This set of experiments was conducted specifically to determine whether centrosome amplification is affected by Stattic and we believe this issue has been addressed. To determine whether centrosome fragmentation is induced by Stattic, we have now counted the number of cells with non-Centrin associated γ -Tubulin staining (new Figure 2d) and did not see a change with Stattic treatment, demonstrating that Stattic does not cause fragmentation.

This is discussed in the text on page 5.

Reviewer Comment:

#16. Page 12-13: The description of PLK4 needs to be clarified. Overexpression of PLK4 is used as a tool for centrosome amplification, as PLK4 regulates of centrosome duplication. There is also no description of what the PLK41-608 is?

Author Response:

PLK4 (1-608) is a truncated version of PLK4 that is catalytically active but cannot dimerize, rendering it unable to induce centrosome amplification. This has now been explained in the text on page 14.

Reviewer #2 (Remarks to the Author):

Considering that many tumor cells harbor excess centrosomes and yet manage to form bipolar spindles by centrosome clustering, the induction of centrosome declustering (which then leads to formation of multipolar spindles and cell death) has emerged as a potentially attractive therapeutic strategy. In this manuscript, Morris, Dedhar and coworkers report a hitherto unexpected role of Stat3 in controlling the extent of centrosome clustering in (tumor) cells displaying excess centrosomes. Their work originates with a screen aimed at identifying small molecule inhibitors of centrosome clustering. From an analysis of substructures of the primary hit compound (KM08165), they zoom in on Stattic, a metabolic derivative of KM08165, that has previously been characterized as an inhibitor of Stat3 (Schust et al. op. cit). Following up on this observation, the authors then search for a pathway through which Stat3 might enhance centrosome clustering. Surprisingly, they find that this novel function of Stat3 does not depend on canonical upstream signaling (i.e. EGF-induced activation of EGFR

or Jak2 kinase), although it does depend on Stat3 dimerisation, nor does it depend on regulation of transcription. They then follow up on a previous report suggesting that Stat3 inhibits the ability of Stathmin to depolymerize microtubules. Having provided evidence that Stattic indeed affects this functionality of Stathmin in vitro, they however claim that the Stat3-Stathmin function in centrosome clustering is largely independent of microtubule depolymerization in vivo. Then, the story takes a somewhat unexpected twist. In an attempt to find downstream effectors of Stat3-Stathmin, the authors survey some 400 protein kinase inhibitors for a possible role in the regulation of centrosome clustering, and this leads them to focus on Plk1. They show that Stattic inhibits Plk1 activity (as measured through analysis of activating Plk1 phosphorylation) and that Plk1 affects centrosome clustering through its (previously established) regulation of gamma-tubulin recruitment. Finally, the authors engineer human breast cancer cells (MDA-MB-231) allowing them to induce centrosome amplification through inducible expression of Plk4, the master regulator of centriole duplication. Using this system they show that the presence of multiple centrosomes sensitizes cells (as well as tumors in NOD/SCID mice) to low doses of Stat3 inhibitors. In conclusion, the authors describe a novel pathway important for the regulation of centrosome clustering. This pathway links Stat3 to Stathmin and Stathmin to Plk1. Their data lead them to propose that Stat3 inhibitors (which are currently in phase III trials) might be particularly effective on tumors showing centrosome amplification.

In my opinion, the paper is very well written: by linking their data to pre-existing “bits and pieces” from the literature, the authors present a “story” that strikes me as intriguing. Overall, I find the data to be of high quality and the interpretation of the results mostly plausible, albeit perhaps not always 100% conclusive. My specific criticism concerns the following points:

Major points:

Reviewer Comment:

1. I have a major issue with the results described in Figure 2c/text on page 5. The authors correctly state that a change in the number of separated “centrosomes” per cell could be due to centrosome declustering, but also to centrosome “amplification” or “fragmentation”. They then use centrin staining to count centriole pairs and use their results to argue against “fragmentation”. The problem with this interpretation is that centrin staining measures centriole numbers and thus constitute a proxy for centrosome amplification – a priori, it says nothing about fragmentation of pericentriolar material. The fact that centriole numbers do not change thus argues against overduplication, not against fragmentation! One common assay for centrosome fragmentation is staining for gamma-tubulin. In fact, fragmentation is typically defined as an increase in gamma-tubulin-positive foci that are able to nucleate microtubules but lack centrioles (and hence lack centrin staining) – because fragmentation only affects ‘pericentriolar material’.

Author Response:

We have now performed this quantification as shown in the new Figure 2d. We did not observe a change in the number of cells with non-Centrin associated γ -Tubulin when Stattic was used, demonstrating that centrosome fragmentation is not induced by Stat3 inhibition.

We have also now added a discussion of these new experiments showing that Stattic does not induce centrosome fragmentation on page 5 of the text.

Reviewer Comment:

2. A second major problem concerns the claim that regulation of centrosome clustering by Stat3 is independent of transcription (Figure 3a,b) and text on page 6. This is a key conclusion of the paper, but I am afraid I am unable to evaluate the validity of this conclusion. In particular, I could not find a clear description of how these experiments were done. A priori I would expect that inhibition of transcription or translation should severely impair cell cycle progression and so I wonder how the authors are able to observe mitoses in apparently undiminished frequency? When describing the experiments aimed at confirming the efficacy of the inhibitors, they report that inhibitors were applied for several hours. Was the same protocol used prior to analysis of mitotic cells? It might help to provide a schematic that illustrates the temporal sequence (and duration) of inhibitor treatments, relative to the time of analysis. For their “story” to stand, it is absolutely critical that the authors exclude a transcriptional effect of Stat3 inhibition on centrosome clustering/declustering.

Author Response:

A schematic describing the timeline in inhibitor experiments is now shown in Supplementary Fig. 1h.

All of the inhibitor based assays of centrosome clustering were carried out in the same way. Cells were treated with the compound for 4 hours then the cells were fixed and stained and scored. The Actinomycin D and Cycloheximide experiments were conducted in this exact same way.

Previous studies have shown that Cycloheximide and Actinomycin D blocks entry into S-phase, slows mitotic progression and blocks completion of cytokinesis. However besides that mitosis proceeds surprisingly unimpaired and these compounds have been used in experiments involving mitotic cells in many papers²⁵⁻²⁹. We found a small decrease in the number of cells in mitosis with Cycloheximide (from 3.6% for untreated cells to 2.9% for 30uM Cycloheximide) as well as for Actinomycin D (from 3.9% for untreated cells to 3.4% for 30uM Actinomycin D). One possible explanation for this relatively small difference is that while cells are being blocked from entering mitosis, they are also being prevented from exiting anaphase and so the two processes roughly equal out. These observations are now discussed in the text on page 7.

We used an established luciferase-based assay to demonstrate that Actinomycin D and Cycloheximide did inhibit transcription/translation, including Stat3-promoter dependent transcription (Figure 3b).

These compounds blocked production of luciferase from both a non-specific promoter (Control Reporter) as well as a Stat3-specific promoter (Stat3 Reporter). The fact that Actinomycin D and Cycloheximide had no effect on centrosome clustering despite inhibiting transcription/translation demonstrates that transcriptional effects are not involved.

Reviewer Comment:

3. Related to the above point, a schematic illustrating the experimental setup used to explore the role of canonical EGF and Jak2 signaling upstream of Stat3 would also be helpful. While I do not consider myself an expert in the Stat3 field, I was under the impression that Stat3 dimerization is controlled by tyrosine kinase receptor activation and downstream Jak2 signaling. If this were correct (but I may be wrong...), how do the authors reconcile the importance of Stat3 dimerization for centrosome clustering with the independence of this Stat3 function from upstream signaling/Jak2?

Author Response:

Cells were treated with Jak2 inhibitor for 4 hours then fixed, stained and scored. This was the same method used for all inhibitor based experiments. A timeline for inhibitor treatment is now shown in Supplementary Figure 1h.

Based on the comments of another reviewer, we are removing reference to Stat3 dimerization because there is controversy whether the Stat3C mutant is in fact a constitutive dimer. We now refer to Stat3C as constitutively active.

The most studied pathway for Stat3 activation does involve Jak2 signalling however many other signals activate Stat3 including NFκB³⁰. Stat3 signalling in mitosis has not been previously explored³⁰ so regulation of Stat3 in mitosis could be completely different from its regulation in interphase. For instance, one of the most prominent features of Stat3 activation is nuclear-cytoplasmic shuffling and Jak2 activation of Stat3 induces Stat3 to accumulate in the nucleus. During mitosis, the nuclear envelope breaks down and therefore nuclear-cytoplasmic shuffling cannot occur. We have added commentary on this to the text on pages 7 and 8.

Reviewer Comment:

4. In Figure 3J, the authors confirm previous literature showing that, in vitro, Stat3 inhibits Stathmin-induced microtubule depolymerization, and that Stat3 inhibitors block this function. To explore the existence of a similar mechanism in vivo, they then ask whether Paclitaxel influences Stathmin-induced centrosome declustering, and because this microtubule stabilizing drug does not produce much of an effect, they conclude that the Stat3-Stathmin function in centrosome clustering is largely independent of microtubule depolymerization. I find this argument rather weak. I would expect that the Stathmin

literature should offer a more direct way to explore the impact of Stattic/Stat3/Stathmin on the stability of microtubule polymers in cells (e.g. using differential extraction)?

Author Response:

As noted by the reviewer, Stat3 knockout has been previously shown to affect the microtubule array of cells³¹. When we partially inhibit Stat3, it is likely that microtubules are slightly different and we do not exclude this. However, the fact that Paclitaxel is unable to rescue centrosome clustering suggests that bulk microtubule disruption is not the cause of Stattic-dependent centrosome declustering. This led us to look at kinase inhibitors which then lead us to PLK1. We believe that the evidence presented for Stat3/Stathmin affecting PLK1 is compelling. We also note that Stathmin has been previously shown to affect PLK1 activity³², giving support to the notion that Stat3/Stathmin could be acting independent of Stathmin depolymerase function.

As the reviewer suggests, differential extraction is a measurement of microtubule stability. If we did see a difference in microtubule stability when we treat with Stat3 inhibitors, we are unclear as to how this would show whether or not this is important for centrosome clustering.

Reviewer Comment:

5. Figure 4 presents several histograms plotting ratios of Western blot signals. Specifically, the authors compare signals obtained with phospho-specific and pan-kinase antibodies, and they show one example of these signals in Figure 4e. Although I recognize that antibodies recognizing specific “activating” phosphorylation sites in kinases can represent valuable tools, these assays are not the most direct or rigorous ways for measuring kinase activities. Thus, while the histograms in Figure 4 collectively suggest a “clear case”, I think the authors should also provide the corresponding primary data (i.e. the corresponding Western blots) in a supplementary Figure.

Author Response:

To clarify, the histograms in Figure 4c were not created from quantifying Western Blots. Rather, this data was generated from “In-Cell Western” assays which are very similar to ELISA-based assays (except using fluorescence instead of luminescence) and are not Western Blots. This assay is performed in 96-well plates and allows quantification of many samples at once. We did perform Western Blots of Phospho-PLK1 using lysates from Stat3-inhibitor treated BT-549 cells (Figure 4e) as well as MDA-MB-231 cells (Supplementary Figure 4b). The In-Cell Western assay is now described in greater detail in the Methods section (Methods Section page 3).

We note that we also showed that Stattic, Stat3 and Stathmin influence PLK1 activity in vitro, using only recombinant proteins and directly measuring PLK1 activity on a model substrate (Figure 4f). This method does not measure Phospho-PLK1 but measures PLK1 kinase activity directly. The fact that we

could reconstitute the entire pathway in vitro is the strongest evidence that the mechanism we propose for signalling can occur.

Reviewer Comment:

6. Figure 4i requires quantification. From this single exemplary IF panel it is impossible to conclude that Myc-Plk1 expressing cells are resistant to the effects of Stattic on centrosome clustering.

Author Response:

We have repeated this experiment on a larger scale and have now quantified the number of Stattic-treated, Myc-PLK1 expressing cells with declustered centrosomes. Myc-PLK1 was expressed in BT549 cells that were treated with 1uM Stattic. This is shown in the new Figure 4i.

Minor points:

Reviewer Comment:

1. IF panels of Figure 5: reading the corresponding text, I would have expected to see a comparison of gamma-tubulin levels in mitotic cells that spontaneously display 2 centrosomes or more than 2 centrosomes (not just cells having undergone drug treatment).

Author Response:

We have changed the DMSO-treated cell in the top panel of Fig. 5a from a cell with 2 centrosomes to a cell with clustered supernumerary centrosomes. We have also changed the text on page 12 to state that we are showing images of cells with supernumerary centrosomes (Fig. 5a, b) and are quantifying both supernumerary and normal bipolar cells (Fig. 5c). These changes make the statements in the text more accurately reflect what is shown in Figure 5.

We note that we quantified γ -Tubulin levels in both 2-centrosome cells as well as supernumerary centrosome cells (Fig. 5c). We did not show images of γ -Tubulin in cells with 2 centrosomes due to space limitations in the Figure.

Reviewer Comment:

2. One comment about “spinning the tale”: having argued against microtubule depolymerization being downstream of Stat3-stathmin, the authors turn to explore a role of kinases – as if this was the most logical next step. However, when reading this passage, their argumentation struck me as less than compelling. My suspicion is that the authors had independently performed kinase inhibitor screens, pointing to Plk1 and Aurora, and later saw an opportunity to link the corresponding data to their work

on Stat3-stathmin. If this were the case, there would be absolutely nothing wrong with this – all good papers present data in a most logical order, not necessarily reflecting history. Nevertheless, I suspect that some re-wording (perhaps a more transparent presentation of the reasons for shifting emphasis to kinases) might smoothen this particular transition in the paper.

Author Response:

We have updated the wording of the text on page 10 to reflect the fact that the kinase screen was independently conducted and that the screen gave us the opportunity to link Stat3-Stathmin to potential downstream signaling.

Reviewer Comment:

3. It might be helpful to briefly “define” MTT assays in the main text.

Author Response:

We have inserted an explanation of the MTT assay in the text on page 14. The steps taken in performing the MTT assay are described in the Methods section on pages 2 and 3.

Reviewer Comment:

4. p. 14: the authors state “This demonstrates that Plk1 inhibition is “uniquely” effective in cells with supernumerary centrosomes” – I think what they mean is that “Plk1 inhibition is “particularly” effective... (“Uniquely” seems the wrong word; after all, Plk1 inhibition is also effective in cells with 2 centrosomes – only less so).

Author Response:

We have corrected this in the manuscript.

Reviewer #3 (Remarks to the Author):

Reviewer Comment:

The use of the “medicinal chemistry” is not correct, because the derivatives were purchased (page 3).

Author Response:

We have removed the term “medicinal chemistry” from the relevant section.

Reviewer Comment:

The structure of the nitro group is not NO₂H, but NO₂. This applies to a total of 4 molecules in Figure 1d and Supplementary Figure 1b.

Author Response:

We have corrected this in the diagrams for both Figure 1d and Supplementary Figure 1b.

Reviewer Comment:

The term “thionaphthene” for the derivative lacking the nitro group is incorrect.

Author Response:

We have corrected this to 1-Benzothiophene 1,1-dioxide in Figures 1d, 1e and 1f.

Reviewer Comment:

The authors use STAT3C as a constitutively active STAT3 mutant and assign this activity to constitutive dimerization. However, PMID 16956893 reports that STAT3C is not a covalent dimer, but instead has elevated activity due to higher DNA binding affinity. The authors might want to comment on this.

Author Response:

Stat3C has been referred to by many papers as the dimeric version but there is clearly controversy whether it is a dimer and what the function of the dimer is. To avoid making unsubstantiated claims, we now refer to Stat3C as constitutively active Stat3 and do not discuss dimerization. This has been corrected in the text on page 8 and in Figures 3e and 4d.

Stat3 function in mitosis has not been previously explored³⁰ and this paper is the first to find a role for Stat3 in mitotic cells. Since we have found that Stat3 transcription factor function is not required for centrosome clustering, we assume that changes in DNA binding are also not relevant. This would be an interesting area to explore in future research though.

References

1. Telentschak, S., Soliwoda, M., Nohroudi, K., Addicks, K. & Klinz, F.J. Cytokinesis failure and successful multipolar mitoses drive aneuploidy in glioblastoma cells. *Oncology reports* **33**, 2001-2008 (2015).
2. Kalatova, B., Jesenska, R., Hlinka, D. & Dudas, M. Tripolar mitosis in human cells and embryos: occurrence, pathophysiology and medical implications. *Acta histochemica* **117**, 111-125 (2015).
3. Silkworth, W.T., Nardi, I.K., Scholl, L.M. & Cimini, D. Multipolar Spindle Pole Coalescence Is a Major Source of Kinetochore Mis-Attachment and Chromosome Mis-Segregation in Cancer Cells. *PLoS one* **4** (2009).
4. Raab, M.S. *et al.* GF-15, a novel inhibitor of centrosomal clustering, suppresses tumor cell growth in vitro and in vivo. *Cancer Res* **72**, 5374-5385 (2012).
5. Chng, W.J. *et al.* The centrosome index is a powerful prognostic marker in myeloma and identifies a cohort of patients that might benefit from aurora kinase inhibition. *Blood* **111**, 1603-1609 (2008).
6. Yu, H., Lee, H., Herrmann, A., Buettner, R. & Jove, R. Revisiting STAT3 signalling in cancer: new and unexpected biological functions. *Nat Rev Cancer* **14**, 736-746 (2014).
7. Pannu, V. *et al.* Centrosome-declustering drugs mediate a two-pronged attack on interphase and mitosis in supercentrosomal cancer cells. *Cell death & disease* **5**, e1538 (2014).
8. Ogden, A. *et al.* Quantitative multi-parametric evaluation of centrosome declustering drugs: centrosome amplification, mitotic phenotype, cell cycle and death. *Cell death & disease* **5**, e1204 (2014).
9. Mittal, K. *et al.* A centrosome clustering protein, KIFC1, predicts aggressive disease course in serous ovarian adenocarcinomas. *Journal of ovarian research* **9**, 17 (2016).
10. Chavali, P.L. *et al.* A CEP215-HSET complex links centrosomes with spindle poles and drives centrosome clustering in cancer. *Nat Commun* **7** (2016).
11. Lee, M.Y., Moreno, C.S. & Saavedra, H.I. E2F activators signal and maintain centrosome amplification in breast cancer cells. *Mol Cell Biol* **34**, 2581-2599 (2014).
12. Peloponese, J.M., Haller, K., Miyazato, A. & Jeang, K.T. Abnormal centrosome amplification in cells through the targeting of Ran-binding protein-1 by the human T cell leukemia virus type-1 Tax oncoprotein. *P Natl Acad Sci USA* **102**, 18974-18979 (2005).
13. Dominguez, D. *et al.* Centrosome aberrations in human mammary epithelial cells driven by cooperative interactions between p16(INK4a) deficiency and telomere-dependent genotoxic stress. *Oncotarget* **6**, 28238-28256 (2015).
14. Pihan, G.A. *et al.* Centrosome defects and genetic instability in malignant tumors. *Cancer Res* **58**, 3974-3985 (1998).
15. Hebert, A.M., DuBoff, B., Casaletto, J.B., Gladden, A.B. & McClatchey, A.I. Merlin/ERM proteins establish cortical asymmetry and centrosome position. *Genes Dev* **26**, 2709-2723 (2012).
16. Miyachika, Y. *et al.* Centrosome amplification in bladder washing cytology specimens is a useful prognostic biomarker for non-muscle invasive bladder cancer. *Cancer Genet-Ny* **206**, 12-18 (2013).
17. Kushner, E.J. *et al.* Excess centrosomes disrupt endothelial cell migration via centrosome scattering. *The Journal of cell biology* **206**, 257-272 (2014).
18. Delaval, B., Bright, A., Lawson, N.D. & Doxsey, S. The cilia protein IFT88 is required for spindle orientation in mitosis. *Nature cell biology* **13**, 461-U262 (2011).
19. Chopra, P., Sethi, G., Dastidar, S.G. & Ray, A. Polo-like kinase inhibitors: an emerging opportunity for cancer therapeutics. *Expert opinion on investigational drugs* **19**, 27-43 (2010).

20. Fan, L. *et al.* Paris saponin VII inhibits metastasis by modulating matrix metalloproteinases in colorectal cancer cells. *Mol Med Rep* **11**, 705-711 (2015).
21. Wang, H.H. *et al.* Clinical and Biological Significance of KISS1 Expression in Prostate Cancer. *Am J Pathol* **180**, 1170-1178 (2012).
22. Ceriani, R.L., Sasaki, M., Sussman, H., Wara, W.M. & Blank, E.W. Circulating Human Mammary Epithelial Antigens in Breast-Cancer. *P Natl Acad Sci-Biol* **79**, 5420-5424 (1982).
23. Akita, N. *et al.* Identification of oligopeptides binding to peritoneal tumors of gastric cancer. *Cancer Sci* **97**, 1075-1081 (2006).
24. Cho, S.C. & Choi, B.Y. Acetylshikonin Inhibits Human Pancreatic PANC-1 Cancer Cell Proliferation by Suppressing the NF-kappaB Activity. *Biomolecules & therapeutics* **23**, 428-433 (2015).
25. Sigoillot, F.D. *et al.* A time-series method for automated measurement of changes in mitotic and interphase duration from time-lapse movies. *PloS one* **6**, e25511 (2011).
26. Liebmann, J., Cook, J.A., Teague, D., Fisher, J. & Mitchell, J.B. Cycloheximide inhibits the cytotoxicity of paclitaxel (Taxol). *Anti-cancer drugs* **5**, 287-292 (1994).
27. Kim, M. *et al.* Caspase-mediated specific cleavage of BubR1 is a determinant of mitotic progression. *Mol Cell Biol* **25**, 9232-9248 (2005).
28. Pathak, S., McGill, M. & Hsu, T.C. Actinomycin D effects on mitosis and chromosomes: sticky chromatids and localized lesions. *Chromosoma* **50**, 79-88 (1975).
29. Guy, A.L. & Taylor, J.H. Actinomycin D inhibits initiation of DNA replication in mammalian cells. *Proc Natl Acad Sci U S A* **75**, 6088-6092 (1978).
30. Sehgal, P.B. Paradigm shifts in the cell biology of STAT signaling. *Semin Cell Dev Biol* **19**, 329-340 (2008).
31. Ng, D.C. *et al.* Stat3 regulates microtubules by antagonizing the depolymerization activity of stathmin. *The Journal of cell biology* **172**, 245-257 (2006).
32. Silva, V.C. & Cassimeris, L. Stathmin and microtubules regulate mitotic entry in HeLa cells by controlling activation of both Aurora kinase A and Plk1. *Molecular biology of the cell* **24**, 3819-3831 (2013).

Reviewers' comments:

Reviewer #1 (Remarks to the Author):

some of my concerns have been addressed by the reviewers. however, i am still unsure to which extent centrosome de-clustering is happening and whether it plays a role in decrease viability upon static treatment.

Original points that have not been convincingly addressed:

#2. In response to this point, the authors mention that centrosome de-clustering causes growth inhibition but not cell death and reference the GF-15 paper. However, this work is highly controversial in the field since GF-15 does not seem to be specific inhibitor of centrosome clustering. When using conditions that prevent clustering, e.g. siRNA against HSET/KIFC1, others have shown that centrosome de-clustering can compromise viability and kill all cells with extra centrosomes. I understand that the authors are choosing a paper that better fits their conclusions but it is not accurate. In addition, the conclusion that cells with extra centrosomes treated with static have defects in clustering is not supported by any data. Without live cell imaging or at least centrosome clustering quantification at telopase/cytokinesis this point cannot be made.

#4. It is really unclear what is killing the cells. The authors seem confused by which extent centrosome de-clustering is affecting viability. From the graphic in fig 1c it is impossible to conclude that cells with extra centrosomes are more sensitive to low doses of the inhibitor because of de-clustering. The expectation, according to the authors own hypothesis, is that low doses of static that do not affect normal cells, affect cells with extra centrosomes. But as it can be seen in 1b, the de-clustering observed in metaphase in BT-549 cells at 1uM is no different from untreated cells and this is the concentration that the authors see the biggest difference in viability in 1c. So are the authors now saying that they cannot conclude that the viability differences in low doses of static are due to de-clustering?

#6. The authors seem to be reluctant in accepting that centrosome markers might not be reliable to quantify centrosome number. It is well accepted in the field that centrosome amplification can only be accurately determined using centriole markers, and not markers for PCM, such as gamma-tub or pericentrin. While these 2 markers are indeed reliable centrosome markers, because they stain the PCM they cannot be used to quantify numbers, which mostly rely on our ability to quantify the number of centrioles. The authors seem to reference lots of papers that use such markers. This is one of the problems with the field. The fact that others use them does not make it acceptable or good. For example the journals such as cell death & disease and journal of ovarian research might not have the same standards related to this issue. It is interesting that the authors mention Chivali et al paper in Nat Comm as an example. In this paper the authors are mostly looking at cells with 2 centrosomes and thus PCM markers to stain the centrosome is only appropriate. However, when looking at centrosome de-clustering in cells with extra centrosomes the authors use a centriole marker, centrin, to properly assess centrosome number and de-clustering.

#7. Same goes for pericentrin. While it is a great centrosome marker, it cannot be used to assess centrosome number or de-clustering.

#8. The authors explanation again does not fit with the expectations. If low doses of static can de-cluster extra centrosomes, then the differences in viability observed between -dox and +dox conditions do not correspond to centrosome amplification shown in 6a. Did the authors quantify centrosome de-clustering in these cells upon static treatment?

#9. The fact that there are no differences in tumour growth regardless of centrosome amplification is assumed by the authors to be due to pharmacokinetics. But it could also be that this is not relevant in vivo. The pre-treatment is a different experiment and does not reflect any sort of drug treatment done in patients. Also the differences even in these conditions are very low. So it is still unclear how relevant this is in terms of treatment.

Reviewer #2 (Remarks to the Author):

The authors have adequately addressed my concerns. I find the paper to be substantially improved and now support publication.

Reviewer #3 (Remarks to the Author):

The depiction of the nitro group in several of the molecules in Figure 1d is unfortunate, because it appears to be bound via oxygen instead of nitrogen.

In Supplementary Figure 1d one oxygen atom of KM08165 has been shifted.

Reviewers' comments

Reviewer #1 (Remarks to the Author):

some of my concerns have been addressed by the reviewers. however, i am still unsure to which extent centrosome de-clustering is happening and whether it plays a role in decrease viability upon static treatment.

Original points that have not been convincingly addressed:

Reviewer Comment

#2. In response to this point, the authors mention that centrosome de-clustering causes growth inhibition but not cell death and reference the GF-15 paper. However, this work is highly controversial in the field since GF-15 does not seem to be specific inhibitor of centrosome clustering. When using conditions that prevent clustering, e.g. siRNA against HSET/KIFC1, others have shown that centrosome de-clustering can compromise viability and kill all cells with extra centrosomes. I understand that the authors are choosing a paper that better fits their conclusions but it is not accurate. In addition, the conclusion that cells with extra centrosomes treated with static have defects in clustering is not supported by any data. Without live cell imaging or at least centrosome clustering quantification at telopase/cytokinesis this point cannot be made.

Author Response

To address concerns of the reviewer whether cells with extra centrosomes have defects in clustering, we have now quantified centrosome clustering in MDA-MB-231-PLK4 cells with and without Static treatment (new Fig. 6b). Static treatment induced centrosome declustering in cells with PLK4-dependent centrosome amplification, however there was not an exact correspondence between centrosome clustering and cell viability. This is now discussed on page 15 of the text.

Also, as suggested by the reviewer, to examine whether centrosome clustering persists in mitosis, we counted the percentage of telophase cells that are multipolar, as has been performed previously¹. This data is presented in the new Supplementary Fig. 5c. We observed a statistically significant increase in the percentage of multipolar telophase cells, demonstrating that declustering persists until telophase/cytokinesis. We have updated the text on pages 15 to discuss this new data.

One overarching concern of the reviewer is that we observed high rates of centrosome declustering but did not observe correspondingly high rates of cell death. Regardless of whether the GF-15 Cancer Research paper is controversial, another paper, in Nature, also provides data in this topic². Ganem et al. (Nature, 2009) closely examined multipolar cell division in MDA-231 cells and observed that 48.9% of the first round of cell divisions produced viable cells (Figure 1c of this paper). When we add in that it has been frequently observed that mitotic cells with declustered centrosomes can recluster their centrosome during telophase³⁻⁵ then the changes in cell viability we observed are entirely consistent

with the published literature. While it may seem surprising, many cancer cell types can manage to survive multipolar cell division. Delayed cell division and not cell death is a commonly observed phenotype.

The only published data we could find that specifically looked at HSET siRNA related changes in cell viability in relation to centrosome amplification was in Figure 7c of Kwon et al (Genes and Development, 2008). In this figure, researchers scored centrosome amplification in different commonly used cell lines and then measured relative cell viability with HSET siRNA treatment. They found that cell lines with frequent centrosome amplification were more sensitive to HSET siRNA and the percentage of cells with centrosome amplification roughly correlates with the percentage loss in cell viability. While this is an interesting finding, the cell lines used do vary in many other ways besides just centrosome amplification and so this result is open to interpretation. The cells could have varied in how resistant they were to declustering and/or apoptosis, how much HSET they expressed, whether redundant pathways were active, how effective siRNA transfections are in the cell line, etc.

Here, we have directly experimentally manipulated centrosome amplification in the same cell line using an inducible system. Since only one variable is changed (centrosome amplification), the method we used is, in our opinion, a more rigorous test of the effects of centrosome clustering inhibitors on cell viability in cells with centrosome amplification.

Reviewer Comment

#4. It is really unclear what is killing the cells. The authors seem confused by which extent centrosome de-clustering is affecting viability. From the graphic in fig 1c it is impossible to conclude that cells with extra centrosomes are more sensitive to low doses of the inhibitor because of de-clustering. The expectation, according to the authors own hypothesis, is that low doses of Stattic that do not affect normal cells, affect cells with extra centrosomes. But as it can be seen in 1b, the de-clustering observed in metaphase in BT-549 cells at 1uM is no different from untreated cells and this is the concentration that the authors see the biggest difference in viability in 1c. So are the authors now saying that they cannot conclude that the viability differences in low doses of Stattic are due to de-clustering?

Author Response

Figures 1b and 1c refer to KM08165 treatments and not Stattic treatments. The data for the Stattic treatment is much more clear than the KM08165 data. As shown in figure 1e and 1f, there is a concentration-dependent correspondence between the effects of Stattic on declustering and viability.

KM08165 is metabolically converted to Stattic over the course of hours (see Supplementary Figure 1g). Since centrosome clustering experiments were always conducted for 4 hours whereas cell viability assays require days, one possible explanation is that there might not have been enough time for sufficient KM08165 to be converted into Stattic in that time.

The primary point of Figure 1 is to demonstrate how we came to the discovery that a Stat3 inhibitor (Stattic) inhibits centrosome clustering and while we agree that the correspondence between KM08165 declustering and effects on viability is less direct, we feel that this correspondence has been clearly shown for Stat3 inhibition (see Figures 1e, 1f and Figure 6 c, d).

Reviewer Comment

#6. The authors seem to be reluctant in accepting that centrosome markers might not be reliable to quantify centrosome number. It is well accepted in the field that centrosome amplification can only be accurately determined using centriole markers, and not markers for PCM, such as gamma-tub or pericentrin. While these 2 markers are indeed reliable centrosome markers, because they stain the PCM they cannot be used to quantify numbers, which mostly rely on our ability to quantify the number of centrioles. The authors seem to reference lots of papers that use such markers. This is one of the problems with the field. The fact that others use them does not make it acceptable or good. For example the journals such as cell death & disease and journal of ovarian research might not have the same standards related to this issue. It is interesting that the authors mention Chivali et al paper in Nat Comm as an example. In this paper the authors are mostly looking at cells with 2 centrosomes and thus PCM markers to stain the centrosome is only appropriate. However, when looking at centrosome de-clustering in cells with extra centrosomes the authors use a centriole marker, centrin, to properly assess centrosome number and de-clustering.

#7. Same goes for pericentrin. While it is a great centrosome marker, it cannot be used to assess centrosome number or de-clustering.

Author Response to Comments #6 and #7

We have now scored centrosome clustering using antibodies against Centrin-2 and Pericentrin at the same time in BT-549 cells (new Supplementary Fig. 2d, e). The scores for declustering were found to be identical which makes us confident that our choice of Pericentrin and Gamma-Tubulin to score centrosome clustering gives reliable data. We have updated the text on page 6 to indicate this. Furthermore, in the few times that Gamma-Tubulin was used to score centrosome clustering, we found very similar results to when using Pericentrin.

Reviewer Comment

#8. The authors explanation again does not fit with the expectations. If low does of stattic can de-cluster extra centrosomes, than the differences in viability observed between -dox and +dox conditions do not correspond to centrosome amplification shown in 6a. Did the authors quantify centrosome de-clustering in these cells upon stattic treatment?

Author Response

We have now quantified centrosome clustering in MDA-MB-231-PLK4 cells with and without Stattic treatment (new Fig. 6b) and have updated the text on page 15 to discuss this new data. Stattic treatment induced centrosome declustering but, as expected, we did not see an exact correspondence between centrosome declustering and cell viability (see Response to Comment #2 for a summary of our position).

Reviewer Comment

#9. The fact that there are no differences in tumour growth regardless of centrosome amplification is assumed by the authors to be due to pharmacokinetics. But it could also be that this is not relevant in vivo. The pre-treatment is a different experiment and does not reflect any sort of drug treatment done in patients. Also the differences even in these conditions are very low. So it is still unclear how relevant this is in terms of treatment.

Author Response

In this paper we have demonstrated that Stat3 regulates centrosome clustering in vitro and in vivo. While most of the experiments involved cells in culture, we have demonstrated that centrosome clustering is indeed significantly inhibited in Stat3 knockout mouse mammary tumours (Fig. 2e). Unfortunately, our attempts to translate these findings utilizing Stattic in vivo did not produce compelling data on tumour growth. Since Stattic also inhibits Stat3 transcription factor function, one would expect some effect on tumour growth, independent of centrosome amplification. We attribute the lack of effect on poor pharmacokinetic properties of Stattic.

Since these data are inconclusive at present and will require more in depth pharmacological studies, we have removed the tumour growth data, as had been suggested by one of the reviewers in the first round of comments.

Reviewer #2 (Remarks to the Author):

The authors have adequately addressed my concerns. I find the paper to be substantially improved and now support publication.

Reviewer #3 (Remarks to the Author):

Reviewer Comment

The depiction of the nitro group in several of the molecules in Figure 1d is unfortunate, because it appears to be bound via oxygen instead of nitrogen.

In Supplementary Figure 1d one oxygen atom of KM08165 has been shifted.

Author Response

We have now changed NO₂ to O₂N in all the relevant models and have removed the misplaced O from Supplementary Figure 1d.

References

1. Konotop, G. *et al.* Pharmacological Inhibition of Centrosome Clustering by Slingshot-Mediated Cofilin Activation and Actin Cortex Destabilization. *Cancer Res* 76, 6690-6700 (2016).
2. Ganem, N.J., Godinho, S.A. & Pellman, D. A mechanism linking extra centrosomes to chromosomal instability. *Nature* 460, 278-282 (2009).
3. Telentschak, S., Soliwoda, M., Nohroudi, K., Addicks, K. & Klinz, F.J. Cytokinesis failure and successful multipolar mitoses drive aneuploidy in glioblastoma cells. *Oncology reports* 33, 2001-2008 (2015).
4. Kalatova, B., Jesenska, R., Hlinka, D. & Dudas, M. Tripolar mitosis in human cells and embryos: occurrence, pathophysiology and medical implications. *Acta histochemica* 117, 111-125 (2015).
5. Silkworth, W.T., Nardi, I.K., Scholl, L.M. & Cimini, D. Multipolar Spindle Pole Coalescence Is a Major Source of Kinetochore Mis-Attachment and Chromosome Mis-Segregation in Cancer Cells. *PloS one* 4 (2009).

REVIEWERS' COMMENTS:

Reviewer #1 (Remarks to the Author):

the authors have addressed all my concerns and the manuscript is now much improved and suitable for publication.